# Genetic evidence that uptake of the fluorescent analog 2NBDG occurs independently of known glucose transporters

**Lucas J. D'Souza**[1], **Stephen H. Wright**[2], **Deepta Bhattacharya**[1]*

**1** Department of Immunobiology, University of Arizona, Tucson, Arizona, United States of America,
**2** Department of Physiology, University of Arizona, Tucson, Arizona, United States of America

* deeptab@arizona.edu

**Data Availability Statement:** RNA-seq data reported in this paper is available at NCBI GEO (accession GSE202181). All other relevant data are

## Abstract

The fluorescent derivative of glucose, 2-Deoxy-2-[(7-nitro-2,1,3-benzoxadiazol-4-yl)-amino]-D-glucose (2NBDG), is a widely used surrogate reagent to visualize glucose uptake in live cells at single cell resolution. Using CRISPR-Cas9 gene editing in 5TGM1 myeloma cells, we demonstrate that ablation of the glucose transporter gene *Slc2a1* abrogates radio-active glucose uptake but has no effect on the magnitude or kinetics of 2NBDG import. Extracellular 2NBDG, but not NBD-fructose was transported by primary plasma cells into the cytoplasm suggesting a specific mechanism that is unlinked from glucose import and that of chemically similar compounds. Neither excess glucose nor pharmacological inhibition of GLUT1 impacted 2NBDG uptake in myeloma cells or primary splenocytes. Genetic ablation of other expressed hexose transporters individually or in combination with one another also had no impact on 2NBDG uptake. Ablation of the genes in the *Slc29* and *Slc35* families of nucleoside and nucleoside sugar transporters also failed to impact 2NBDG import. Thus, cellular uptake of 2NBDG is not necessarily a faithful indicator of glucose transport and is promoted by an unknown mechanism.

## Introduction

Glucose is a critical nutrient for fueling cellular metabolism. After its import, glucose can be catabolized through a multitude of metabolic pathways. For example, glucose can be broken down by glycolysis into pyruvate that in turn is oxidized to fuel the tricarboxylic acid (TCA) cycle and generate ATP [1]. Glucose is also important for generating intermediate sugars which serve as precursors for the synthesis of nucleotides, fatty acids, and glycosylation sugars [2]. Glucose uptake and catabolism are key defining features of cells, distinguishing between developmental stages of lineages, tumors versus normal tissues, and activation status.

Definitive methods to measure glucose import have classically involved the use of isotope-labelled derivatives of glucose [3–5]. A shortcoming to these compounds, however, is their rapid breakdown or export from the cell, and an inability to resolve glucose uptake at single cell resolution. Fluorescent derivatives of glucose, which allow for visualization and flow

within the manuscript and its Supporting Information files.

**Funding:** This work was supported by NIH grant R01AI129945 (D.B.). L. D. was supported by a Bio5 Postdoctoral fellowship award. The use of the Imagestream was made possible by the NIH award S10 OD028466. The funders had no role in study design, data collection and analysis, decision to publish, or preparation of the manuscript. There was no additional external funding received for this study.

**Competing interests:** D.B. is a co-founder of Clade Therapeutics. Sana Biotechnology has licensed intellectual property of D.B. and Washington University. Gilead has licensed intellectual property of D.B. and Stanford University. S.H.W and L.D. report no conflicts of interest.

cytometric estimation of glucose uptake in cells, can potentially overcome these problems [6]. One such widely used compound is 2-deoxy-2-(7-Nitro-2,1,3-benzoxadiazol-4-yl)amino-D-glucose (2NBDG), in which the 2-hydroxyl group of D-glucose is replaced with a fluorescent 7-Nitrobenzofurazan group [7]. This compound was first characterized in *E. coli* where it competed with D-glucose for import via a mannose or a glucose/mannose transporter system [7, 8]. Since its discovery, it has been used across many mammalian cell types and *in vitro* culture models as a surrogate for glucose uptake. In plasma cells, we used this analog to demonstrate that 2NBDG+ long-lived plasma cells showed elevated spare respiratory capacity relative to their 2NBDG- short-lived counterparts, thereby linking glucose uptake with plasma cell longevity [9, 10]. Further, as compared to other markers of murine plasma cells, 2NBDG positivity correlated well with the longevity of the plasma cell subset [11]. Yet despite its widespread use, 2NBDG uptake has not been definitively shown to occur through glucose transporters.

Glucose import into eukaryotic cells can take place via three families of transporters: the sodium-glucose linked symporters of the SGLT/SLC5 family of transporters; the SWEET family of glucose transporters of the SLC50 family; and lastly, the well characterized GLUT/SLC2 family of sugar transporters [12, 13]. Through much of B cell development and activation, glucose uptake is mediated by the *Slc2* family member GLUT1, which is thus likely to be the chief glucose transporter in plasma cells [14]. Our assumption had been that GLUT1 would be the likeliest candidate transporter for 2NBDG uptake in plasma cells as well. Contrary to our expectations, we demonstrate in this study that disruption of GLUT1 expression led to loss of glucose import but had no effect on 2NBDG uptake. Ablation of other candidate transporters also did not affect 2NBDG uptake. These data are consistent with three very recent pharmacological and genetic studies [15–17]. We conclude that 2NBDG is transported independently of known glucose transporters and thus should not be used to estimate glucose uptake by mammalian cells.

## Materials and methods

### Ethics statement

All animal procedures carried out in this manuscript were approved and carried out based on guidelines provided by the Institutional Animal Care and Use committee at The University of Arizona (approval 17–266). Euthanasia was performed by administering carbon dioxide at a rate of 1.5L/minute in a 7L chamber until 1 minute after respiration ceased. Mice were then cervically dislocated to ensure death.

### Mice

C57BL/6N mice were purchased from the Charles River laboratories and housed under specific pathogen free conditions. Experiments were carried out on sex-matched mice between 8–12 weeks of age.

### Primary cells and cell lines

The mouse myeloma line 5TGM1 was a gift from Michael H. Tomasson at the Washington University in St. Louis [18]. They were maintained in a T-25 flask at 37°C with 5% $CO_2$ and split every 4 days at a ratio of 1:10. Cas9-expressing 5TGM1 cells were generated by spin infecting $2 \times 10^6$ 5TGM1 cells with lentiCas9-BLAST lentivirus and 8μg/mL of Polybrene (Millipore Sigma) at 2500rpm for 90 minutes followed by selection in 10μg/mL Blasticidin-S-HCl (Gibco). Guide RNA (gRNA) containing lentiviruses were introduced into these cells by similar spin infections, followed by selection with 10μg/mL Puromycin (Gibco) at 48 hours.

Alternatively, 5TGM1 cells were spin-infected with lentiCRISPRv2-mCherry lentivirus to generate a Cas9-expressing line that showed mCherry fluorescence. These cells were purified from uninfected cells by fluorescence-activated cell sorting. LentiX-293T cells (632180, Takara Bio) were used for lentivirus packaging and assembly. Cells were cultured in a 100mm petri dish and split using 0.05% Trypsin-EDTA (Millipore Sigma) when it exhibited greater than 80% confluence. Single cell suspensions from spleens and bone marrows of mice were prepared and erythrocytes lysed with an ammonium chloride-potassium (ACK) lysis buffer. Lymphocytes were then isolated using a Histopaque-1119 (Millipore Sigma) and cells were suspended in 1x PBS containing 5% adult bovine serum (FACS buffer).

## Chemicals and cell culture media

2NBDG (11046) and 1NBDF (9002314) were both purchased from Cayman chemical. These reagents were resuspended at a final concentration of 10mg/mL in 1xPBS and in some cases diluted further to 1mg/mL in 1xPBS. Both reagents were completely soluble at both concentrations and did not require a solubilizing agent. Mice were injected intravenously with 100µg of either 2NBDG or 1NBDF and euthanized after 20 minutes. To measure 2NBDG uptake *in vitro*, $1 \times 10^6$ 5TGM1 cells were cultured with 20µg/mL (~60µM) 2NBDG (Cayman Chemical) in complete media for 1 hour at 37˚C followed by antibody staining unless otherwise indicated. 5TGM1 cells were cultured in RPMI (Gibco) containing 10% fetal bovine serum (PEAK Serum), 2mM L-alanyl-L-glutamine, 1mM sodium pyruvate, minimal non-essential amino acids, penicillin, and streptomycin (all from Corning). Glucose-free RPMI (11879–020, Gibco) was used for some experiments, and extraneous D-glucose (G5767, Millipore Sigma) was supplemented to the media at the indicated final concentrations. Lenti-X 293T cells were maintained in DMEM (11965, Gibco) containing 10% fetal bovine serum, L-alanyl-L-glutamine, minimal non-essential amino acids, sodium pyruvate, penicillin, and streptomycin. For assays in sodium-free media, minimum essential medium was used in which sodium salts were replaced by an equal quantity of potassium salts. GLUT1 inhibitors cytochalasin B, BAY-876, and WZB-117 (C6762, SML1774, and SML0621 respectively, all from Millipore Sigma) were all dissolved in DMSO and used at their indicated concentrations. Carbenoxolone (C4790, Millipore Sigma) was dissolved in sterile water and diluted to its final concentration in complete RPMI. For some experiments, 4-Chloro-7-Nitrobenzofurazan (A14165, Thermo Fisher Scientific) was diluted in methanol and used.

## Plasmids and lentivirus generation

lentiCas9-Blast, lentiCRISPRv2-mCherry, and lentiGuide-puro were all used in this study (52962, 52961, and 52963 respectively; Addgene). gRNA sequences were chosen from the existing mouse Brie library or designed using the CRISPick platform and cloned into lentiGuide-puro as described previously [19–21]. In brief, 20-mer gRNA sequences were introduced into primers (Millipore Sigma) and after hybridization were cloned into Esp3I-digested lentiGuide-puro using the NEBuilder® HiFi DNA Assembly Master mix (E2621, NEB). Successful constructs were verified in plasmid from isolated colonies by sequencing with the LKO.1 5' primer. Constructs used for generation of the *Slc2a3*, *Slc2a5*, *Slc2a6*, and *Slc2a8* knockout cell line were prepared by digesting lentiGuide-puro constructs with already cloned gRNA sequences and control lentiGuide-puro with BsiWI-HF and MluI-HF (R0553 and R3198, NEB) to exclude the puromycin N-acetyltransferase gene. Sequences encoding fluorescent proteins were then cloned into these digested vectors to generate *Slc2a3*-lentiGuide-dsRed, *Slc2a5*-lentiGuide-mMaroon, *Slc2a6*-lentiGuide-mCherry, and *Slc2a8*-lentiGuide-tagBFP along with their respective no gRNA controls.

LentiX-293T cells (632180, Takara Bio) were cultured to 60% confluence in $10cm^2$ dishes and transfected using 30μL GeneJuice Transfection reagent (Millipore Sigma), 1.75μg of pMD2.G (12259, Addgene), 3.25μg of psPAX2 (12260, Addgene), and 5μg of the lentiviral construct as per the manufacturer's instructions. Media was changed at 6 hours post transfection and supernatants collected at 48 and 72 hours and filtered through a 0.45μm syringe. Filtered supernatants were then mixed in a 5:1 ratio with 25% polyethylene glycol-8000 (Millipore Sigma) in 1xPBS and incubated overnight at 4˚C. Samples were then spun down at 2500rpm for 20 minutes and pellets resuspended in 100μL 1xPBS. Aliquots were then frozen at -80˚C until use.

## Flow cytometry

All fluorescence associated cell sorting was carried out on a FACS Aria II and analysis on a BD LSR II or a BD Fortessa (Becton Dickinson). The following anti-mouse antibodies were used: CD138-Phycoerythrin (PE), -Allophycocyanin (APC), or -BV510 (281–2, Biolegend) and B220-BV421 (RA3-6B2, Biolegend). When staining cultured cells, propidium iodide (Millipore Sigma) or Zombie UV (Biolegend) were used to exclude dead cells from the analysis. To stain for GLUT1, cells were first fixed in 2% paraformaldehyde (Electron Microscopy Services), permeabilized in 0.1% Saponin (84510, Millipore Sigma) and then stained with an unconjugated GLUT1 monoclonal antibody (SPM498, Thermo Fisher Scientific) followed by a Rat anti-mouse IgG2a -Alexa Fluor 647 detection antibody (SB84a, Southern Biotech). Cells stained with the detection antibody alone were used as an isotype control. Data was analyzed using the FlowJo software (Becton Dickinson). To monitor 2NBDG uptake kinetics, $1x10^6$ cells were resuspended in the appropriate buffer and 2NBDG introduced to the suspension just prior to recording the sample. Mean Fluorescence Intensity (MFI) changes over time were monitored using the Kinetics platform on FlowJo.

## Imaging flow cytometry

2NBDG treated 5TGM1 cells expressing mCherry with control or *Slc2a1*-targeting gRNA were stained for surface CD138. Samples were then analyzed on an Imagestream[X] Mk II (Luminex) and 3000 events recorded per group at 60x magnification. In separate experiments, expression of GLUT1 in deleted cultures were verified by staining cells with CD138 and GLUT1 as described in the previous section. Hoechst 33342 (62249, Thermo Fisher Scientific) was used to identify the nucleus in some assays. Raw information files analyzed on the IDEAS v.6.3 software (Luminex) and similarity morphology indices calculated using the nuclear localization wizard. Mean similarity morphology indices for groups across three experiments were then graphed using Prism 9.2 (Graphpad).

## [14]C-Glucose uptake assays

$1x10^6$ 5TGM1 cells were resuspended in glucose-free incomplete RPMI (Gibco) in a microcentrifuge tube and incubated briefly in a water bath set at 37˚C. [14]C Glucose (275 mCi/mmol; NEC042X050UC, Perkin Elmer) was introduced into these cultures to reach a final concentration of 0.5 μCi/mL and cells were allowed to incubate with shaking for 30 minutes. Cells were then washed once with 1x PBS (HyClone) and then lysed in a solution containing freshly prepared 0.5N NaOH and 1% SDS for 30 minutes at room temperature. After neutralization with 1N HCl, cell lysate was loaded onto a 96-well plate (Grenier-BioOne) and mixed with MICRO-SCINT[TM]-20 scintillation fluid (Perkin Elmer). Samples were then incubated at room temperature for 2 hours and incorporated radioactivity analyzed on a 1450 MicroBeta TriLux Microplate Scintillation and Luminescence counter (Perkin Elmer).

## Next generation sequencing

Genomic DNA was isolated from gRNA-transduced cultures at the time of assay using a DNA extraction kit (IBI Scientific). Primers were designed +/-150bp from the PAM site of the targeting gRNA and used to amplify 300bp amplicons from 1μg genomic DNA using the Q5® DNA polymerase (M0491, NEB) for 25 cycles. Amplicons were then gel extracted, amplified with the same primers and reaction conditions for 2 cycles, and purified by gel extraction. Equimolar concentrations of amplicons from different reactions were then pooled and 300fmol of the sample was end-prep treated and ligated with adapter sequences according to the ligation sequencing kit protocol (SQK-LSK109, Oxford Nanopore). For some libraries, barcodes were introduced into the samples using a modified version of the ligation sequencing kit protocol (NBD-104, Oxford Nanopore). Samples were then loaded onto primed SpotON flow cells and sequenced on a MinION Mk1c (Oxford Nanopore) with a read filter of 250-500bp and high-accuracy basecalling. Reads in resultant FASTQ files were mapped to the region of interest and indels calculated using the CRISPResso2 analysis pipeline [22]. Unmodified, in-frame, and frame-shift mutation containing read frequencies were graphed as stacked columns using Prism 9.2 (Graphpad).

## RNA sequencing

Total RNA was extracted from control gRNA-, *Slc2a1* gRNA-, or *Slc2a1/3/5/6/8* gRNA-transduced 5TGM1-Cas9 cells using the Nucleospin RNA XS kit (Takara) according to the manufacturer's instructions. cDNA was generated using oligo-dT primers, fragmented, and Illumina primers ligated for sequencing by the Novogene Corporation Inc. Paired 150bp reads were then sequenced on a NovaSeq 6000 (Illumina). FASTQ files containing 48–69 million reads were then mapped using Salmon and differential gene expression analysis carried out using DESeq2 [23, 24]. All analysis was done by uploading FASTQ files onto the Galaxy public platform, https://usegalaxy.org [25]. Data was plotted on Prism 9.2 (Graphpad). The accession number for the RNA-seq data reported in this paper is NCBI GEO GSE202181.

## Statistical analysis

All statistical analysis was carried out on Prism 9.2 (Graphpad). Specific tests used and significance are indicated in the figures and accompanying figure legends. Adjusted p-values and fold changes for RNA-Seq data were calculated on DESeq2 [24].

# Results

## GLUT1 does not mediate uptake of 2NBDG

The glucose transporter GLUT1, encoded by the gene *Slc2a1*, is highly expressed by multiple myeloma cells, which are transformed counterparts of long-lived plasma cells [26]. Using lentiviral transduction, we engineered 5TGM1 mouse myeloma cells to constitutively express the Cas9 protein. Using lentiviruses, we then expressed four different guide RNAs (gRNAs) that targeted exons 3, 4, or 5 in the *Slc2a1* locus in these 5TGM1-Cas9 cells. The frequency of GLUT1-positive cells in these cultures dropped to approximately 50% relative to that of a control gRNA-transduced culture (Fig 1A). Consistent with a previous report, we observed GLUT1 at the plasma membrane as determined by colocalization with the surface marker CD138 in GLUT1-sufficient cells(S1 Fig) [27]. As expected, GLUT1-negative cells in these same *Slc2a1*-deleted cultures, however, showed complete loss of this transporter on both the surface and in the cytosol (S1 Fig). This loss in expression of GLUT1 was accompanied by an approximate 80% decrease in $^{14}$C-glucose uptake (Fig 1B). Some of the GLUT1-positive cells

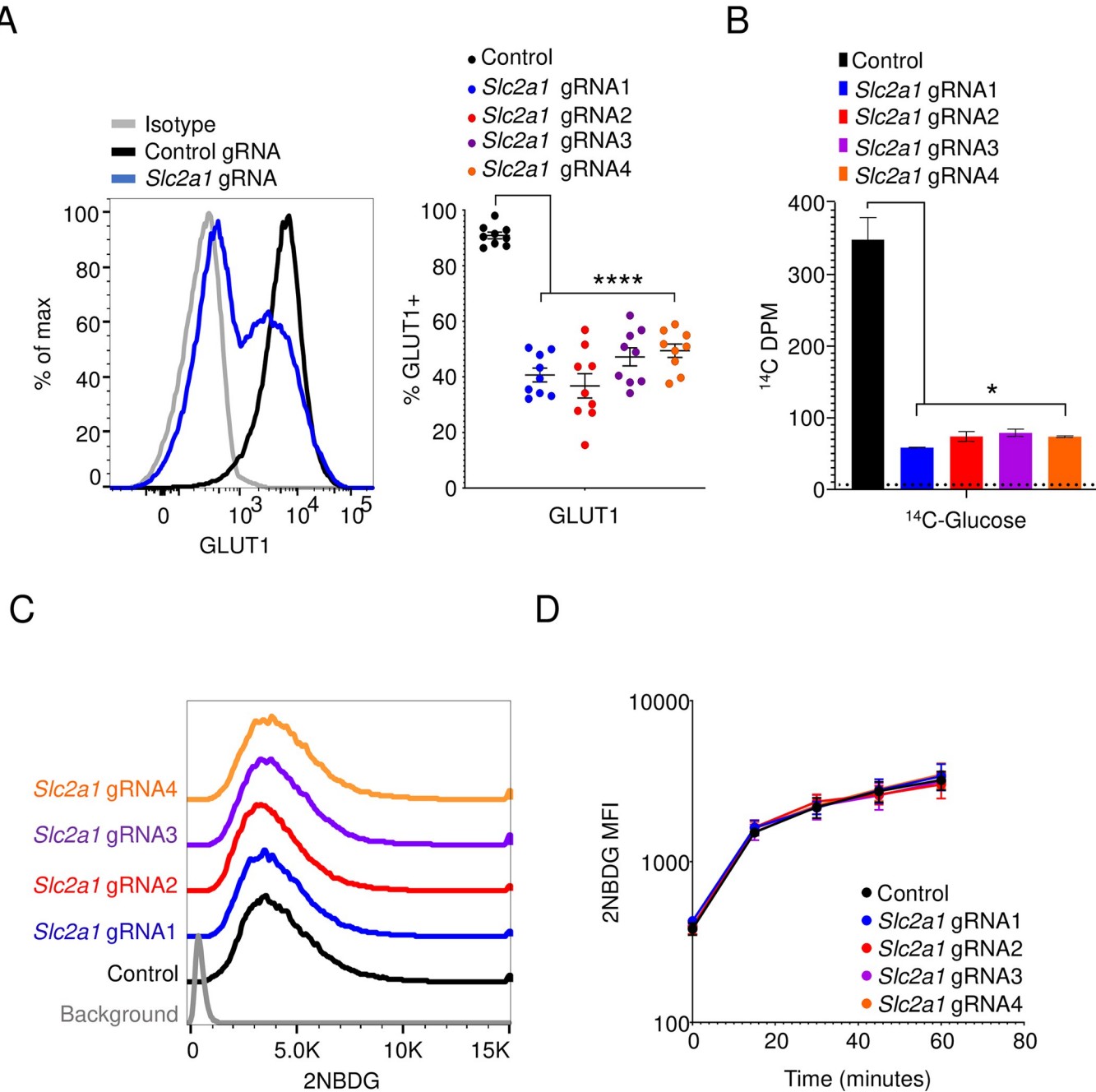

**Fig 1. 2NBDG uptake is unaffected by *Slc2a1* deletion in 5TGM1 cells.** (A) GLUT1 staining of gRNA-transduced cells. Representative histogram (left) showing GLUT1 expression on 5TGM1 cells transduced with a control gRNA (black) and a *Slc2a1*-targeting gRNA (blue). Quantification of percent GLUT1 positive cells (right) in control and *Slc2a1* gRNA-transduced cells across 9 independent experiments. Each circle represents a single group from a single experiment. ***p<0.0001 by Brown-Forsyth and Welch one-way ANOVA multiple comparisons test. (B) $^{14}$C-glucose uptake in gRNA-transduced cells. Lysates from cells transduced with gRNAs as in (A) were cultured in $^{14}$C-glucose containing media for 30 minutes and $^{14}$C signal was quantified in the cell pellet. Background counts indicated by the dotted line. Representative graph of three independent experiments. *p<0.05 by Brown-Forsyth and Welch one-way ANOVA multiple comparisons test. (C) Flow cytometric analysis of 2NBDG uptake in control gRNA- (black) and *Slc2a1* gRNA-transduced (colored) 5TGM1-Cas9 cells. Cells were cultured in media containing 2NBDG for 60 minutes. Histogram representative of three independent experiments. (D) 2NBDG uptake in control gRNA (black) and *Slc2a1* gRNA-transduced (colored) 5TGM1-Cas9 cells. Mean fluorescence intensity (MFI) +/- SEM shown for 0-, 15-, 30-, 45-, and 60-minutes post culturing with 2NBDG. Pooled data from three independent experiments. No significant differences were observed with ordinary two-way ANOVA with Dunnett's multiple comparison test.

in the deleted cultures may represent cells with one functional copy of GLUT1 and as a result show a partial reduction in [14]C-glucose uptake. Further, [14]C-glucose can be catabolized into various cellular metabolites, indicating a simultaneous measure of uptake and metabolism in this assay. Put together, this might explain the slight discrepancy between the degree of reduction in GLUT1-positive cells and the extent of [14]C-glucose uptake in deleted cultures. To our surprise, however, 2NBDG import was unaffected in *Slc2a1*-targeted gRNA cultures (Fig 1C). Further, we saw no change in the kinetics of 2NBDG uptake in these cells, measured over the course of an hour (Fig 1D). These findings suggest that mechanisms or transporters other than GLUT1 can mediate uptake of 2NBDG.

## Plasma cells take up 2NBDG, but not 1-NBD-Fructose

We next explored other possible mechanisms by which cells take up 2NBDG. Solute entry into cells can take place via three main pathways: diffusion, endocytosis-mediated uptake, or import through solute carriers and other transporters. We considered the possibility that 2NBDG, which is hydrophilic, is endocytosed by cells in a non-specific manner rather than delivered to the cytoplasm via a transporter. To test this, we examined 5TGM1 cells treated with 2NBDG using imaging flow cytometry and observed that 2NBDG was distributed evenly across the cytosol of cells (Fig 2A). Weak colocalization was observed with the nuclear stain Hoechst 33342 and no colocalization with the surface marker CD138 (Fig 2A). No punctate localization was observed as one would expect from vesicle-mediated uptake or endocytosis (Fig 2A). The absence of the sugar transporter GLUT1 did not affect this distribution, confirming a *Slc2a1*-independent mechanism of 2NBDG uptake (Fig 2B).

We next examined the specificity of 2NBDG uptake in plasma cells. We injected mice with 2NBDG or 1-NBD-fructose (1NBDF), a derivative of fructose in which the NBD moiety is attached to the C-1 sugar of fructose (S2 Fig) [28]. We observed 2NBDG uptake in both splenic and bone marrow plasma cells, with the latter showing a higher frequency of 2NBDG-positive cells (Fig 3A), as we previously reported [9, 10]. Plasma cells, however, did not detectably import 1NBDF (Fig 3A). These data argue against endocytic fluid phase accumulation of 2NBDG, as this mechanism would be predicted to also lead to the accumulation of the chemically similar 1NBDF. We also measured uptake of 4-Chloro-7-nitrobenzofurazan (4C7NB), the fluorescent substrate used in the synthesis of 2NBDG. Unlike 2NBDG, this compound is hydrophobic and showed a linear increase in MFI over time in 5TGM1 cells (S3 Fig). 5TGM1 cells treated with 2NBDG, however, reached a steady state intensity in less than 30 seconds and maintained it for the remainder of the assay (S3 Fig). This pattern of kinetics of 2NBDG accumulation is indicative of transporter-mediated uptake, as under conditions of diffusion or endocytosis, uptake is a linear function of time. We observed similar 2NBDG uptake kinetics in primary mouse plasma cells, suggesting that the mechanism of 2NBDG transport is similar between the two cell types (Fig 3B). Further, 2NBDG import was higher in plasma cells as compared to B cells and total spleen cells (Fig 3B). Put together, these data suggest that plasma cells specifically import 2NBDG through a transporter(s) other than GLUT1.

## GLUT1 inhibition or ablation does not affect 2NBDG uptake kinetics

To further assess the role of GLUT1 in 2NBDG uptake, we pharmacologically inhibited GLUT1 by treating cells with cytochalasin B (CytoB), a well-established GLUT1 inhibitor [29, 30]. As CytoB also affects actin polymerization, we tested 2NBDG uptake in the presence of more specific GLUT1 inhibitors, namely BAY-876 and WZB-117 [31–33]. We found no change in 2NBDG intensity in drug-treated groups compared to untreated or DMSO-treated cells (Fig 4A). We also tested the role of GAP junctions and hemidesmosomes in 2NBDG import by treating cells with

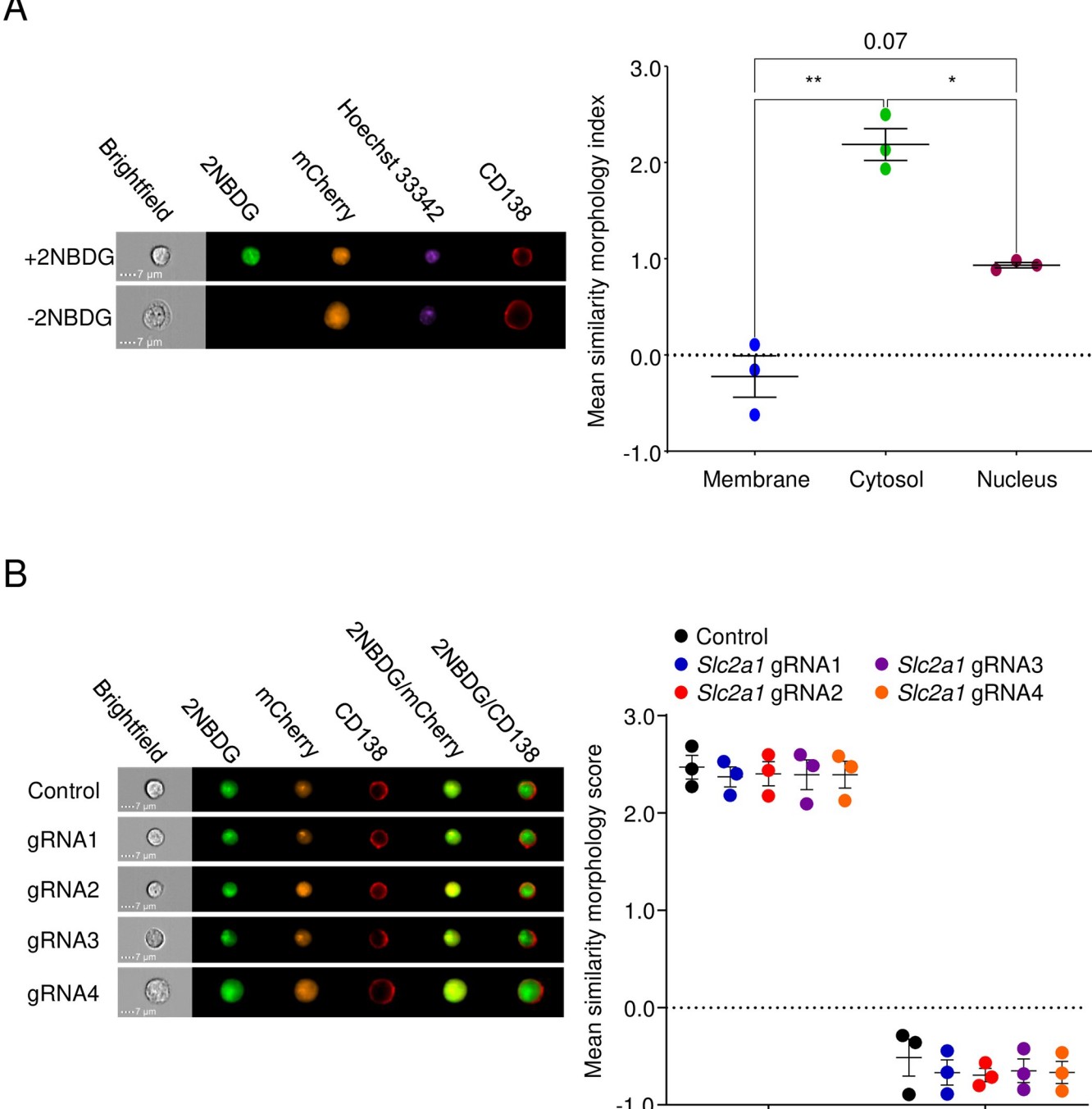

**Fig 2. Plasma cells take up 2NBDG and retain it in the cytosol.** (A) 2NBDG is retained in the cytosol of cells. 5TGM1 cells previously transduced with a Cas9-T2A-mCherry lentivirus were treated with 2NBDG, stained for surface CD138 and Hoechst 33342, and analyzed by imaging flow cytometry. Representative images of cells treated with 2NBDG (top left) relative to untreated cells (bottom left) at 60x magnification are shown. Mean similarity morphology indices for 2NBDG and other stains/dyes were quantified and shown (right). (B) Cas9-mCherry expressing 5TGM1 cells were transduced with control gRNA or *Slc2a1* gRNA lentiviruses. Representative images of each gRNA group (left) at 60x magnification are shown with overlays for 2NBDG/CD138 and 2NBDG/mCherry. Mean similarity morphology indices were quantified (right). For both panels, each dot indicates the mean value for a group in a single experiment. No significant differences were observed with ordinary two-way ANOVA with Dunnett's multiple comparison test.

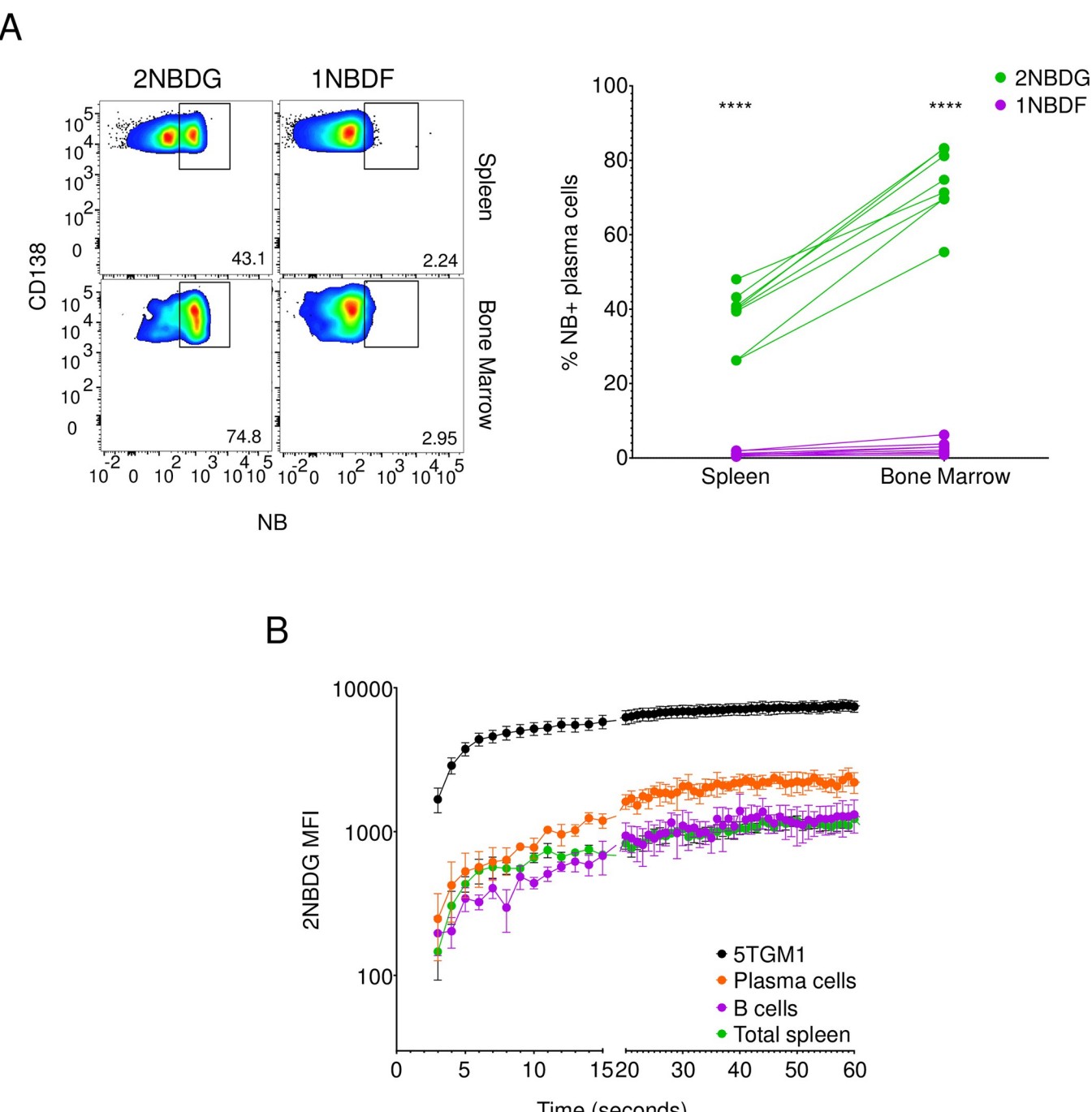

**Fig 3. Plasma cells specifically transport 2NBDG.** (A) Mice were injected with 100μg of either 2NBDG or 1NBDF and assessed for NB fluorescence in splenic and bone marrow plasma cells. Representative flow cytometry plots (left) on splenic (top row) and bone marrow (bottom row) CD138+ plasma cells showing gated percent NB-positive cells. Quantification of NB-positive plasma cell percentages (left) in the spleen and bone marrow and groups from each mouse are connected by a line. Data from three independent experiments with n = 8 mice in both groups. ****p<0.0001 by ordinary two-way ANOVA with post hoc Šídák's multiple comparison test. (B) Freshly isolated *ex vivo* plasma cells (CD138+ B220+/-, orange), B cells (CD138- B220+, purple), and total spleen cells (green) as well as cultured 5TGM1 cells (black) were examined for 2NBDG uptake by flow cytometry. Pooled data from three independent experiments shown as mean+/-SEM for mentioned time points.

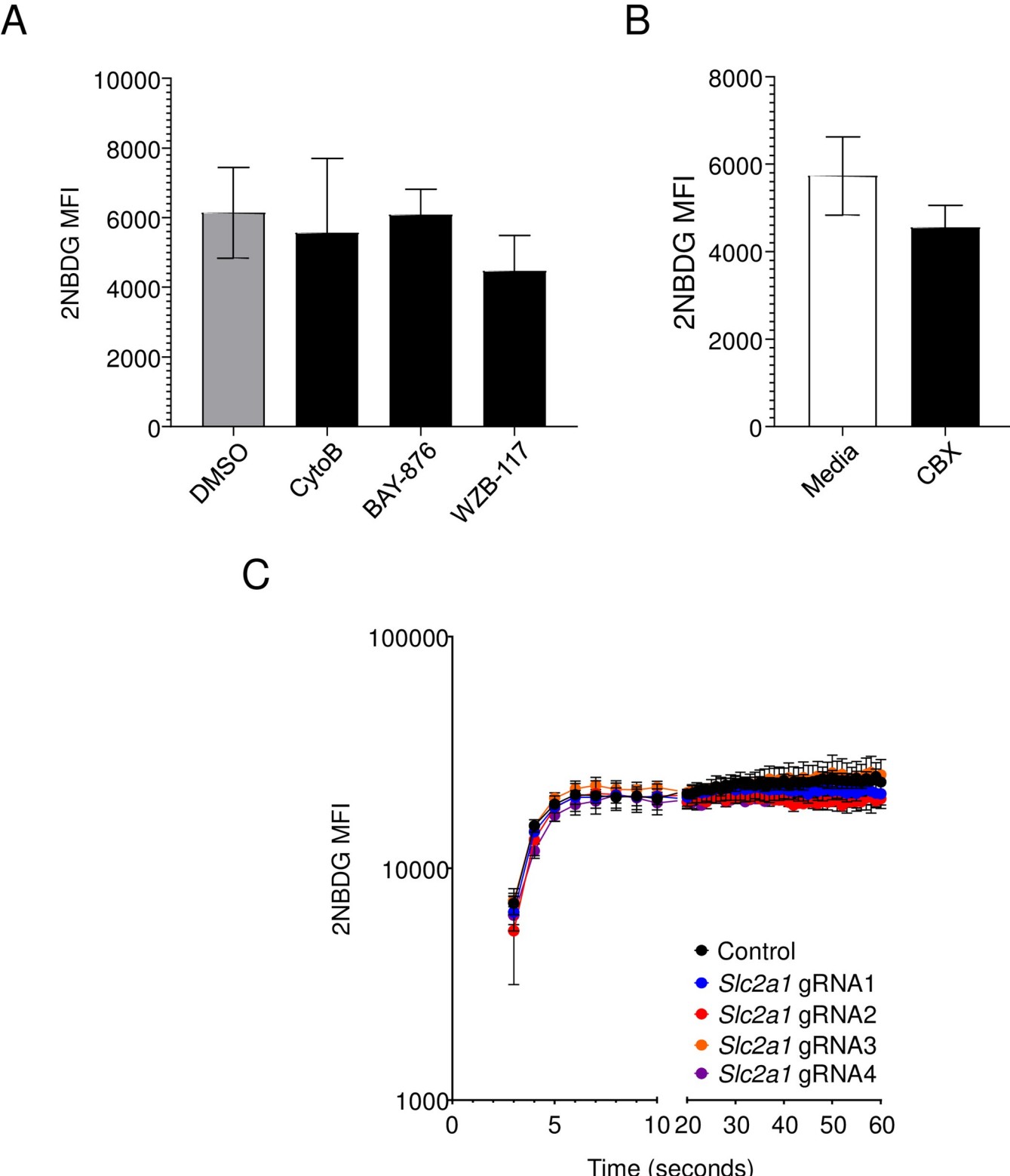

**Fig 4. The kinetics of 2NBDG uptake is unaffected by GLUT1 inhibition or deletion.** (A) 5TGM1 cells were treated with DMSO, 30µM Cytochalasin B (CytoB), 10µM BAY-876, or 100µM WZB-117 for 30 minutes in complete media followed by 60µM of 2NBDG for another 30 minutes. Mean +/- SEM of 2NBDG MFI are shown as bar graphs. Pooled data from three independent experiments. No significance was observed by one-way ANOVA. (B) 5TGM1 cells were treated with 300nM Carbenoxolone (CBX) and assayed for 2NBDG uptake as in (A). Mean +/- SEM of 2NBDG MFI are shown as bar graphs. Pooled data

from three independent experiments. No significance observed by the paired t-test. (C) Control gRNA- or *Slc2a1* gRNA-transduced 5TGM1-Cas9 cells were administered 2NBDG at a final concentration of 60μM and intensity monitored immediately after addition by flow cytometry. Mean +/- SEM is shown for each group for the mentioned time points. Pooled data from three independent experiments.

the blocker Carbenoxolone (CBX) [33–35]. We found no effect of the compound on 2NBDG uptake (Fig 4B). Further, genetic ablation of *Slc2a1* in 5TGM1 cells did not affect 2NBDG import in 5TGM1 cells at any of the time points assayed (Fig 4C). Put together, these findings strongly suggest that 2NBDG transport is independent of GLUT1 and glucose uptake.

## 2NBDG uptake does not depend on other glucose transporters

We next hypothesized that 2NBDG uptake into cells could take place via other transporters implicated in glucose uptake. We performed RNA-Seq on 5TGM1 cells and compared the gene expression profiles with published gene expression data on 5TGM1 cells and primary plasma cells to identify candidate glucose and 2NBDG transporters of the *Slc2*, *Slc5*, and *Slc50* families [9, 10, 36]. Of the 13 members of the *Slc2* family in mice, primary plasma cells and/or myeloma cells expressed SLC2A3, SLC2A6, and SLC2A8 as candidate glucose transporters in addition to SLC2A1 (Fig 5A). Data from another group also identified SLC2A5 in addition to the aforementioned SLC2 transporters [36]. Expression of SLC5 family members was not detected. We also observed high levels of expression of SLC50A1, though this transporter is primarily involved in glucose efflux rather than import [13].

To test if these transporters are responsible for 2NBDG import, we transduced gRNAs targeting these genes into 5TGM1-Cas9 cells and measured 2NBDG uptake by flow cytometry. Sequencing of gRNA targets in these cultures demonstrated 31–50% frameshift mutations, confirming efficient ablation of the intended transporters (Fig 5B). Disruption of these genes, however, did not affect 2NBDG uptake (Fig 5C). Moreover, the frequency of 2NBDG-negative cells in all cultures was similar to those seen in control gRNA-transduced cultures (Fig 5D).

One possible explanation for the unaltered 2NBDG uptake in the *Slc2a1*, *Slc2a3*, *Slc2a5*, *Slc2a6*, and *Slc2a8* deleted cultures is functional redundancy between these sugar transporters. As a result, loss of one of these genes might be compensated by activity of other transporters. To test this possibility, we generated 5TGM1 cells carrying gRNAs targeting all the expressed members of the SLC2 family. Because GLUT1-deficient cells show reduced viability, we first generated a cell line that was ablated for *Slc2a3*, *Slc2a5*, *Slc2a6*, and *Slc2a8* using lentiviral transduction followed by fluorescence activated cell sorting of the reporter positive cells. After sorting and expansion of this line, we then introduced lentiviruses expressing a gRNA targeting *Slc2a1* and examined for 2NBDG uptake 4 days after transduction. Analysis of *Slc2a1* sequences in these cultures showed an average of 50% frameshift mutations, while the other *Slc2* members had 54–65% frameshifts in their respective sequences (Fig 6A). Yet 2NBDG uptake was equivalent in these cells relative to controls (Fig 6B–6D).

We considered the possibility that *Slc2a1*-deleted 5TGM1 cells upregulate other members of the SLC2 family as a compensatory mechanism. Relative to control gRNA-transduced cells, we observe a statistically significant reduction in SLC2A1 and SLC50A1 transcript levels in the deleted cultures, but no noticeable change in the transcript levels of any of the other members of the SLC2 and SLC5 families (S4 Fig). Thus, no known glucose transporter is involved in 2NBDG uptake.

## Exogenous glucose does not inhibit 2NBDG uptake

We next hypothesized that 2NBDG uptake may be mediated by an unidentified glucose transporter. In this case, 2NBDG import into cells would be impeded by competing amounts of D-

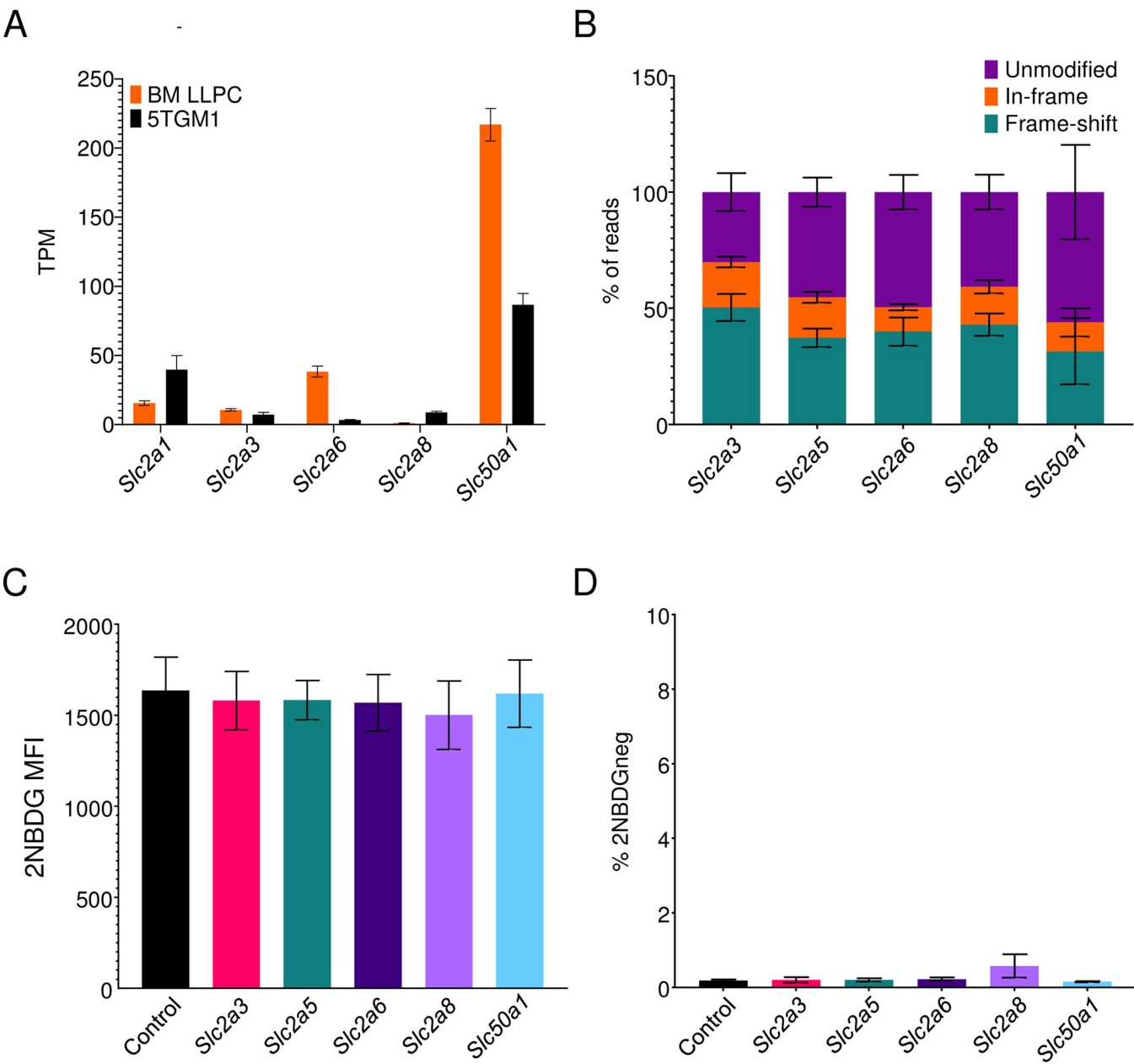

**Fig 5. 2NBDG uptake takes place independently of sugar transporters.** (A) Transcripts per million kilobase (TPM) values of sugar transporters in *ex vivo* bone marrow plasma cells (orange) and 5TGM1 cells (black). Mean values +/- SEM are shown. (B) Quantification of gene modifications in gRNA-transduced cultures. Exons of indicated genes were PCR amplified and sequenced to assay for in-frame and frame shift mutations. Mean values +/- SEM shown for each of the genes and modifications within it. Pooled data from three experiments. (C) 2NBDG uptake in sugar transporter deleted cultures. 5TGM1-Cas9 cells were transduced with control gRNA (black) or gRNAs targeting *Slc2a3* (pink), *Slc2a5* (blue), *Slc2a6* (violet), *Slc2a8* (purple), and *Slc50a1* (cyan). MFIs across three independent experiments are quantified and displayed with SEM. No significant differences observed with Brown-Forsyth and Welch one-way ANOVA multiple comparisons test. (D) Frequency of 2NBDG-negative cells for groups in (C). 2NBDG- gate drawn based on control cells cultured without 2NBDG in complete media. Mean values +/- SEM are shown. No significant differences observed with Brown-Forsyth and Welch one-way ANOVA multiple comparisons test.

glucose in the culture medium. To test this, we quantified 2NBDG uptake in 5TGM1 cells suspended in glucose-free RPMI-1640 or with media containing increasing amounts of D-glucose, reaching 500-fold excess relative to 2NBDG concentrations in the media. By flow

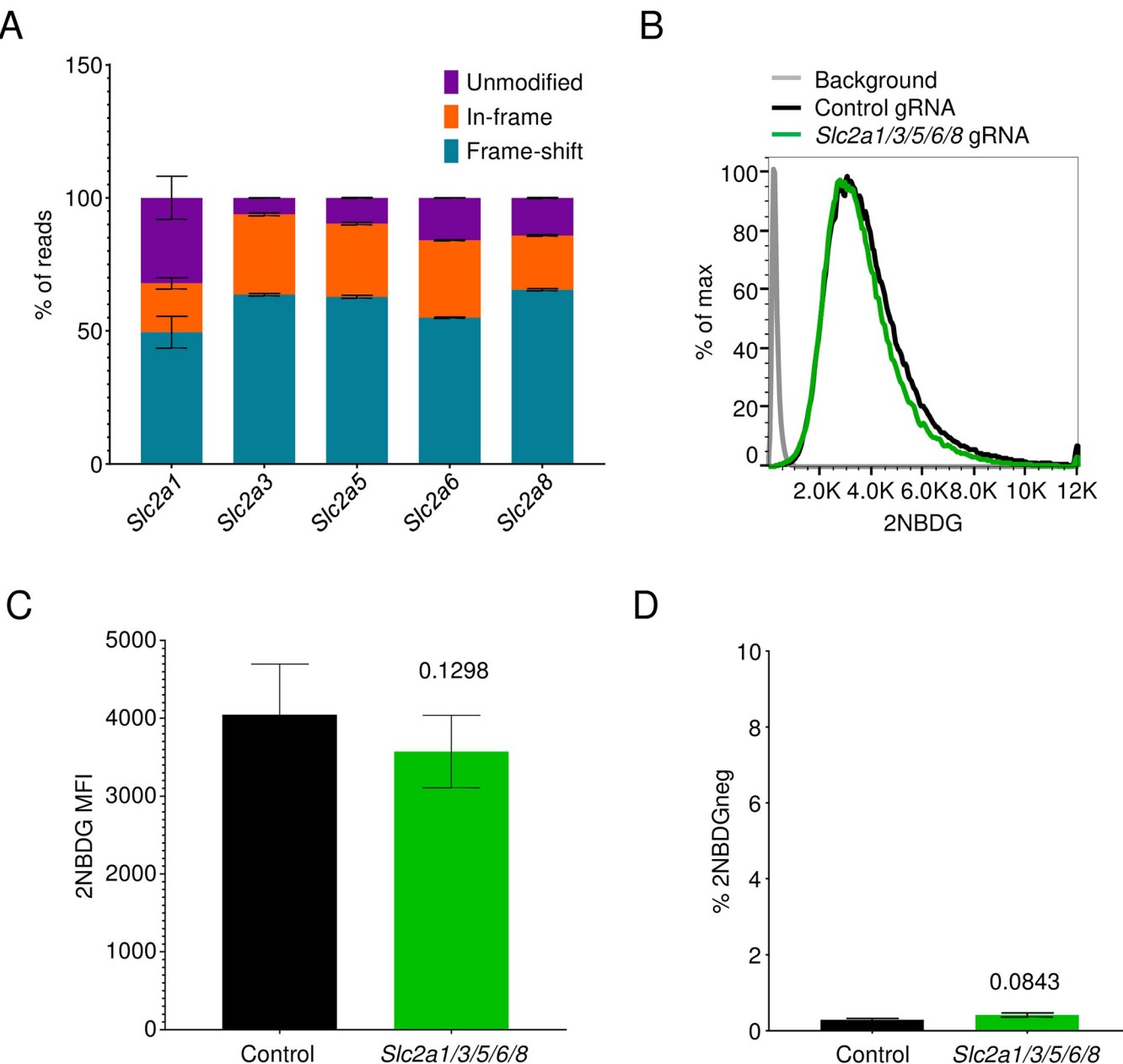

**Fig 6. Combined deletion of sugar transporters does not affect 2NBDG uptake.** (A) Quantification of in-frame and frameshift mutations in *Slc2a1*, *Slc2a3*, *Slc2a5*, *Slc2a6*, and *Slc2a8* genes of the *Slc2a1/3/5/6/8* deleted cultures. Pooled data from three experiments. (B) 2NBDG uptake in *Slc2a1/3/5/6/8* cultures. Representative histogram (left) showing control gRNA- (black) and the five gRNA-transduced cultures (green). Pooled MFI +/- SEM for both groups across three independent experiments are depicted (right). No significance observed with the paired t-test. (C) Frequencies of 2NBDG-negative cells in cultures described in (B). 2NBDG- gating carried out as in Fig 5D. Mean values +/- SEM are shown for both groups. No significance observed with the paired t-test.

cytometry, we observed 2NBDG uptake to reach a steady state in all groups rapidly post-addition but found no difference in the mean 2NBDG intensity in cells under glucose-sufficient or -deficient conditions (Fig 7A). We observed a similar trend in *ex vivo* primary plasma cells, where if anything, cells showed higher 2NBDG intensity in the presence of D-glucose (Fig 7B). Competing glucose in the assay media did not impact 2NBDG intensities in splenic B cells or total spleen cells at any point in the assay (Fig 7C and 7D). Put together, these findings indicate that 2NBDG import is independent of glucose uptake and transporters.

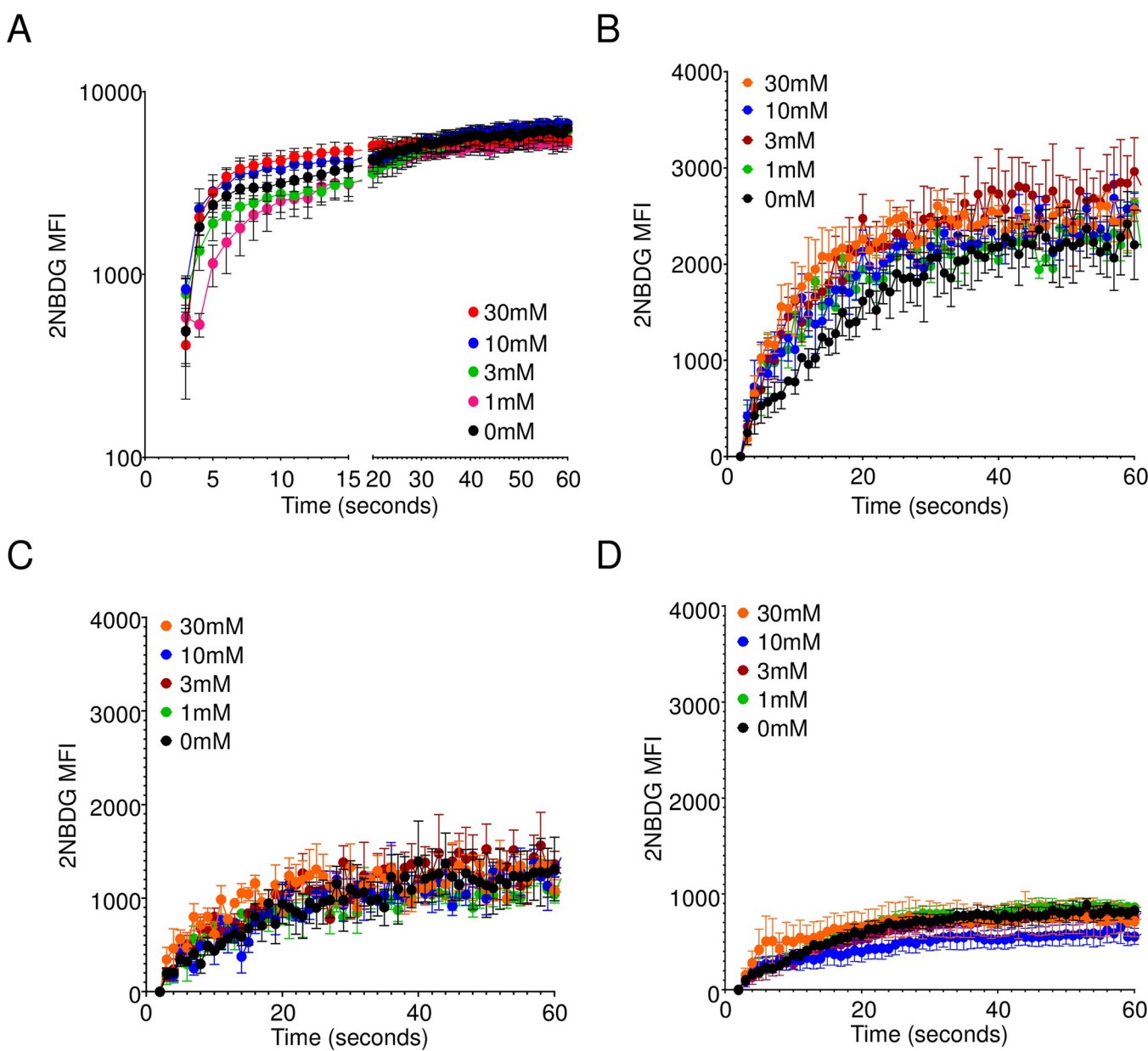

**Fig 7. 2NBDG uptake in cells is independent of competing glucose.** (A) 5TGM1 cells were resuspended in either glucose-free RPMI or RPMI with D-glucose at the mentioned final concentrations. 2NBDG was added at a final concentration of 60μM and intensity monitored immediately after addition by flow cytometry. (B-D) CD138+ enriched spleen cells were resuspended in glucose-free RPMI or RPMI with D-glucose at the specified concentrations. 2NBDG was added and intensity monitored immediately by flow cytometry. 2NBDG intensity in (B) plasma cells (CD138+ B220+/-), (C) B cells (CD138- B220+), and (D) unenriched total spleen cells are shown for the mentioned time points. Mean +/- SEM shown for each group for the indicated time points. Pooled data from three independent experiments each.

## Nucleoside and nucleoside-sugar transporters do not transport 2NBDG

Given the utility of 2NBDG import as a marker of plasma cell longevity, we performed experiments to test other candidate transporters. The structure of 2NBDG mimics nucleotides and nucleotide sugars, which are imported through the *Slc29* and *Slc35* families of transporters, respectively (S2 Fig). RNA-seq data showed *Slc29a1* and *Slc29a3* expression in both primary

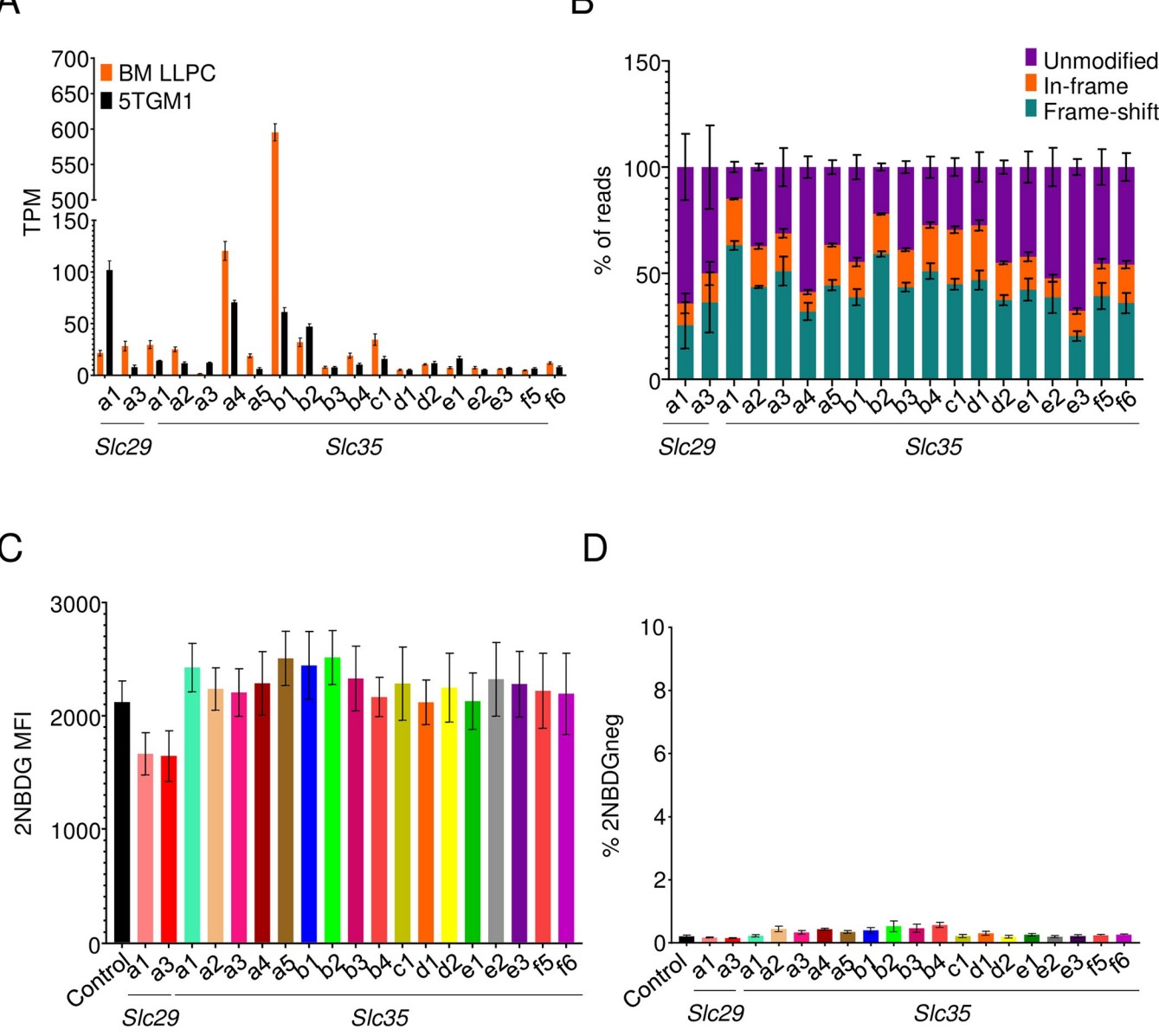

**Fig 8. 2NBDG is not imported through nucleoside or nucleoside-sugar transporters.** (A) TPM values of nucleoside and nucleoside-sugar transporters in *ex vivo* bone marrow (orange) plasma cells and 5TGM1 cells (black). Mean values +/- SEM are shown. (B) Estimation of indels in nucleoside and nucleoside-sugar transporter deleted cultures done as in Fig 5B. Mean values +/- SEM shown for each of the genes and modifications within it. Pooled data from three independent experiments. (C) 2NBDG uptake in deleted cultures. 5TGM1-Cas9 cells transduced with gRNAs targeting indicated nucleoside or nucleoside-sugar transporters (colored) or with a control gRNA (black). 2NBDG MFIs across three independent experiments are quantified and displayed with SEM. (D) Frequencies of 2NBDG negative cells in cultures described in (C). Pooled data from three experiments. No statistical significance observed with the Brown-Forsyth and Welch one-way ANOVA multiple comparisons test.

bone marrow plasma cells and 5TGM1 cells (Fig 8A). Moreover, 17 of the 27 known *Slc35* members in mice were also expressed (Fig 8A). CRISPR-Cas9 ablation of these transporters led to 20–59% frameshift mutations in sequences from 5TGM1 cultures (Fig 8B). However, none of these mutations affected 2NBDG uptake (Fig 8C and 8D). Thus, the *Slc29* or *Slc35* families of nucleotide and nucleotide sugar transporters are not required for 2NBDG uptake.

### 2NBDG uptake is a low-affinity, sodium independent process

To further understand the kinetics of 2NBDG uptake in cells, we next examined 2NBDG import in 5TGM1 cells at differing concentrations of 2NBDG. This would enable us to assess saturability of the compound and estimate parameters like the Michaelis-Menten constant (Km) and Vmax, thereby narrowing down a list of candidate transporters. We treated 5TGM1 cells with dilutions of 2NBDG, starting at a maximum concentration of 6mM and titrating down to 6μM, the lowest dose at which we could detect fluorescence above background. We found that 2NBDG steady state intensity was reached rapidly and maximum intensity titrated nearly proportional to the concentration of 2NBDG (Fig 9A). We did not observe saturation at any of the concentrations we tested, thereby precluding calculations for Km and Vmax. We also examined 2NBDG transport in the absence of exogenous sodium ions, an important cofactor for multiple symporter and antiporter systems in the cell, including the SGLT/SLC5 family of transporters [37]. Cells in sodium free media, however, showed a higher 2NBDG intensity at all time points assayed (Fig 9B). We may infer from these findings that 2NBDG uptake is possibly a low affinity process and does not require sodium ions to mediate its uptake. Further, while specific, 2NBDG uptake is not mediated by known sugar, nucleotide, or nucleotide sugar transporters.

## Discussion

Fluorescent derivatives of glucose like 2NBDG have been used as a tool for visualizing glucose uptake in cells and *in vivo*. However, its import through mammalian sugar transporters has not been demonstrated genetically. Using CRISPR-Cas9 editing of myeloma cells, we show that the sugar transporter GLUT1 is important for glucose uptake but has no role in 2NBDG transport in these cells. Although glucose can be imported into cells via other sugar transporters, we also found no role for these alternative routes in 2NBDG transport. Finally, we found that excess glucose had no impact on 2NBDG uptake in primary splenocytes. As such, our findings show a disconnect between glucose uptake and 2NBDG transport in mammalian cells.

During the course of this work, three reports were published by independent groups that arrived at similar conclusions. The first report by Sinclair *et al.* demonstrated that double-positive (DP) thymocytes took up very little glucose as compared to activated CD8+ T cells in culture [15]. The 2NBDG uptake by cells in these cultures, however, showed the exact opposite trend and was unaffected by pharmacological inhibition of glucose transporters [15]. The second group took advantage of the L929 fibroblast line which expresses GLUT1 as its sole glucose transporter [38]. Using pharmacological inhibitors, siRNA mediated knockdowns, and GLUT1 overexpression, the authors were able to modulate glucose uptake in these cells, but not of 2NBDG or another fluorescent derivative of glucose, 6NBDG [16]. Another recent study observed a lack of *in vivo* correlation between uptake of $^{18}$F-fluorodeoxyglucose, a well-verified reporter of glucose uptake, and 2NBDG [17]. Our findings provide an independent genetic confirmation for each of these reports, and further extend the disparity between glucose and 2NBDG uptake to other glucose transporters within the *Slc2* and *Slc50* families. The use of CRISPR-Cas9 to induce gene deletions in our system offers a robust means to provide this independent confirmation.

In plasma cells, 2NBDG uptake marks longer lived subsets and identifying its transporter could provide insights into mechanisms regulating the longevity of these cells. Long-lived plasma cells do possess more glucose-dependent spare respiratory capacity than do short-lived plasma cells [10]. However, our original conclusions that long-lived plasma cells import more glucose than their short-lived counterparts will need to be revisited. The as-yet unidentified

A

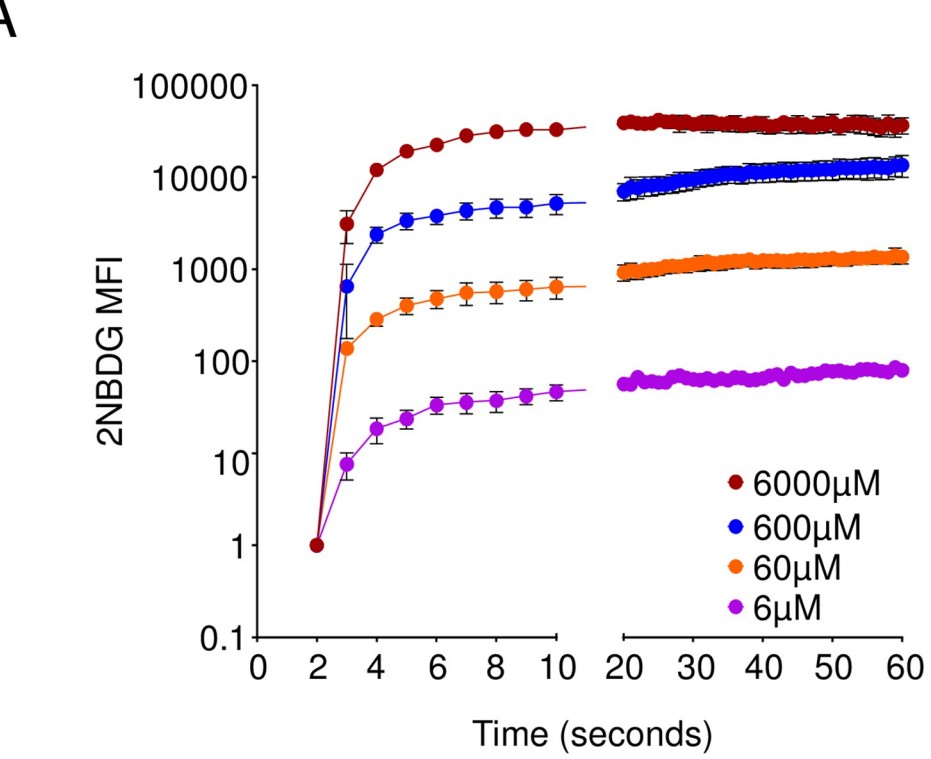

B

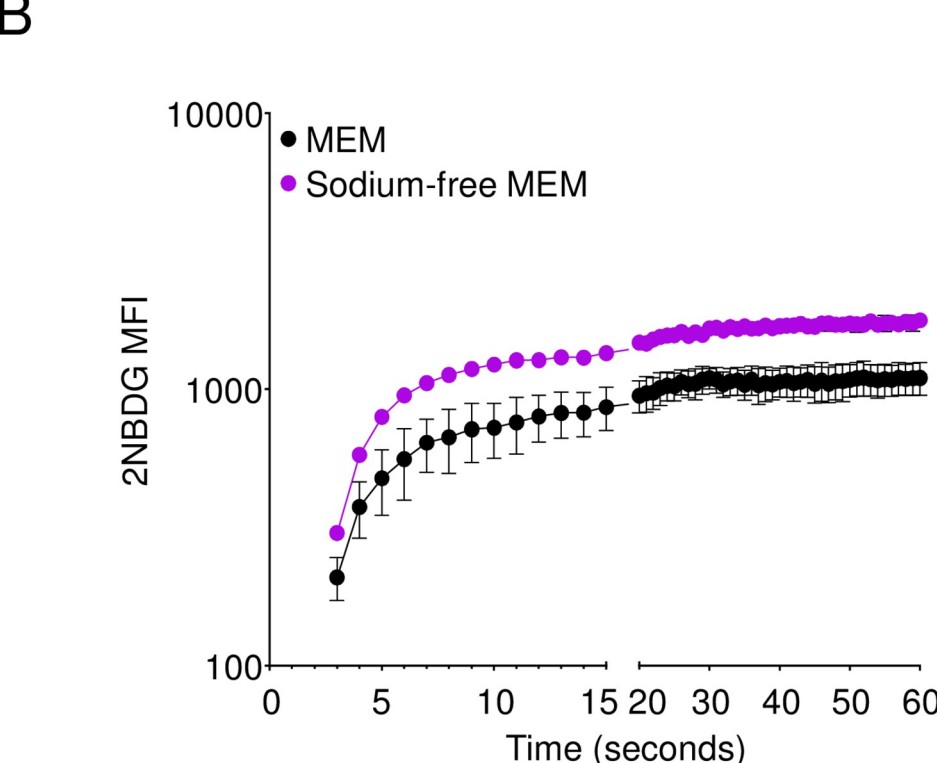

**Fig 9. 2NBDG transport is a low affinity process.** (A) 2NBDG was added to 5TGM1 cells resuspended in 1xPBS to the specified final concentrations and uptake measured by flow cytometry. (B) 5TGM1 cells were resuspended in either DMEM or Sodium-free MEM and measured for 2NBDG import by flow cytometry. Mean +/- SEM shown for each group for the indicated time points. Pooled data from three independent experiments each.

transporter seems specific, as our data indicate that cells deposit 2NBDG in the cytosol. The kinetics of 2NBDG uptake would indicate it is mediated through a transporter, albeit through a low-affinity process. We attempted an unbiased genome-wide gRNA library screen in the 5TGM1 cell line but were unable to identify any known candidates in sorted 2NBDG-negative cultures. The lack of any putative targets in the screen suggests that 2NBDG might be imported into cells via multiple functionally redundant transporters. Such transporters could potentially be identified through an overexpression library screen in cells that intrinsically do not take up 2NBDG. As the mechanisms of 2NBDG uptake remain unclear, we strongly advise that despite its convenience, it should not be considered as a direct indicator of glucose uptake.

## Supporting information

**S1 Fig. *Slc2a1* deletion leads to a loss of surface GLUT1 expression.** Cells transduced with *Slc2a1* gRNA were stained for GLUT1 and CD138. (A) Representative images from GLUT1-sufficient (top) and GLUT1-deficient cells (bottom) in the same culture. (B) Quantification of mean similarity morphology indices for GLUT1 and CD138 in GLUT1+ (filled bars) and GLUT1- (hollow bars) cells. Pooled data from three independent experiments. *p<0.05 by Šídák's multiple comparisons test.
(TIF)

**S2 Fig. Structures of 2NBDG and 1NBDF in relation to glucose and naturally occurring nucleosides.** Chemical structures of (A) D-Glucose, (B) 2NBDG, (C) 1NBDF, (D) 4C7NB, (E) Guanosine, (F) Adenosine, (G) Deoxythymidine, (H) Uridine, and (I) Cytosine. Structures generated using ChemDraw v.20.1.1.
(TIF)

**S3 Fig. 2NBDG and 4C7NB show different uptake kinetics in 5TGM1 cells. 5TGM1 cells were treated with 60μM 2NBDG (black) or 1μM 4C7NB (orange) and the intensity of each compound was monitored over time by flow cytometry. Pooled data from three independent experiments.**
(TIF)

**S4 Fig. *Slc2* deletion does not lead to compensatory expression of other transporters.** RNA-Seq analysis of sugar transporter transcript levels in control gRNA-transduced (black), *Slc2a1* gRNA-transduced (blue), and *Slc2a1/3/5/6/8* gRNA-transduced (green) 5TGM1-Cas9 cultures. Three biological replicates were analyzed for each population and data is represented as mean values +/- SEM. *p<0.05, **p<0.01, and ***p<0.001 by paired two-way ANOVA.
(TIF)

## Acknowledgments

We wish to thank the flow cytometry core at the University of Arizona for their assistance with flow cytometry and the Nikolich-Žugich laboratory for the use of the BD Fortessa for some experiments. We are also grateful to Tyler J. Ripperger for his assistance with RNA-Seq.

## Author Contributions

**Conceptualization:** Lucas J. D'Souza, Stephen H. Wright, Deepta Bhattacharya.

**Formal analysis:** Lucas J. D'Souza, Deepta Bhattacharya.

**Funding acquisition:** Deepta Bhattacharya.

**Investigation:** Lucas J. D'Souza, Deepta Bhattacharya.

**Methodology:** Lucas J. D'Souza, Deepta Bhattacharya.

**Project administration:** Deepta Bhattacharya.

**Visualization:** Lucas J. D'Souza.

**Writing – original draft:** Lucas J. D'Souza.

**Writing – review & editing:** Lucas J. D'Souza, Stephen H. Wright, Deepta Bhattacharya.

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
