## [Decision Letter · Decision Letter 0]

8 Feb 2022

PONE-D-21-38751Genetic evidence that uptake of the fluorescent analog 2NBDG occurs independently of known glucose transportersPLOS ONE

Dear Dr. Bhattacharya,

Thank you for submitting your manuscript to PLOS ONE. After careful consideration, we feel that it has merit but does not fully meet PLOS ONE’s publication criteria as it currently stands. Therefore, we invite you to submit a revised version of the manuscript that addresses the points raised during the review process.

Your paper was reviewed by four experts. As they point out, your claim is very strong and needs more data to support it. For example, use of other cell types and kinetic analyses must be performed. Please revise your manuscript according to the comments by the reviewers and carefully specify your claims / conclusions based on the data in your hand.

We look forward to receiving your revised manuscript.

Kind regards,

Hodaka Fujii, M.D., Ph.D.

Academic Editor

PLOS ONE

Journal Requirements:

(This work was supported by NIH grant R01AI129945 (D.B.). L. D. was supported by a Bio5 Postdoctoral fellowship award. The use of the Imagestream was made possible by the NIH award S10 OD028466. he funders had no role in study design, data collection and analysis, decision to publish, or preparation of the manuscript)

Reviewers' comments:

Reviewer's Responses to Questions

**Comments to the Author**

1. Is the manuscript technically sound, and do the data support the conclusions?

Reviewer #1: Partly

Reviewer #2: Yes

Reviewer #3: Partly

Reviewer #4: Partly

2. Has the statistical analysis been performed appropriately and rigorously? 

Reviewer #1: Yes

Reviewer #2: Yes

Reviewer #3: No

Reviewer #4: Yes

3. Have the authors made all data underlying the findings in their manuscript fully available?

Reviewer #1: Yes

Reviewer #2: Yes

Reviewer #3: No

Reviewer #4: Yes

4. Is the manuscript presented in an intelligible fashion and written in standard English?

Reviewer #1: Yes

Reviewer #2: Yes

Reviewer #3: Yes

Reviewer #4: Yes

5. Review Comments to the Author

Reviewer #1: In this paper, the authors present data regarding the effect of CRISPR-mediated deletion of several classes of membrane transport proteins on 2NBDG uptake in primary plasma and 5TGM1 myeloma cell lines. The authors observe continued 2NBDG uptake despite significant reduction in glucose transport with knockdown of the primary GLUT transporter (SLC2A1) in these cells. To search for possible explanations for this unexpected finding, they perform a fairly extensive, though not exhaustive, set of experiments to knock down other transporters, including one experiment where they simultaneously knock down each of the GLUT isoforms detected in the 5TGM3 cells. While overall the data agree with recent reports using other methodologies that 2NBDG may not serve as a reliable indicator of overall glucose uptake and utilization in mammalian cells, this manuscript has several limitations, based largely on the tools utilized and the experimental conditions chosen, that require either modification of the conclusions reached or addition of additional data. In particular, while the identity of the novel putative transporter(s) may not be clear, within the context of this paper it can and should be directly established whether or not sustained 2NBDG import is due to a carrier-mediated process.

Specific comments and critique:

1. In the paper abstract, the authors claim that CRISP-mediated knockout of Slc2a1 alters the kinetics of 2NBDG uptake. However, the paper does not directly assess transport kinetics. The only data shown is a time course for mean fluorescence intensity. The scale shown is relatively long and does not allow assessment of zero-trans uptake. To make this claim, much more data needs to be shown including classic experiments to assess for transport-mediated uptake. This includes demonstration of saturability and at a minimum estimation of the Km and Vmax of the transport process.

2. In Figure 1, there is a discrepancy in the degree of reduction of GLUT1 positive cells (~50%) and the degree of 14C glucose uptake (~80%) that is not adequately addressed in the manuscript text. This may be due in part to the experimental conditions used to assess for glucose transport activity. What is shown is the combined effect in GLUT1 expressing and non-expression cells with 30-minute incubation with radiolabeled substrate. Under these conditions, there are influences of both uptake and metabolism of glucose. It is recommended that assays be performed at shorter time points and with non-metabolizable substrate (e.g. 2-deoxyglucose). At a minimum, the authors need to show (or adequately reference) the kinetics of glucose transport activity in these cell lines under the conditions used.

3. The interpretation of the data shown in Figure 2 is overstated. In these experiments, the authors seek to establish that the uptake of 2NBDG does not occur via a non-specific endocytosis-mediated process. Demonstration of uniform cytosolic staining of 2NBDG does not prove that this is mediated through a transport-mediated process. Furthermore, the comparison to 1NBDF, while showing that there is specificity for 2NBDG over the other substrate (implying an effect on glucose over fructose uptake), it does not directly follow that these data is not from GLUT1 mediated transport.

4. Although the authors use RNA-seq to assess the specific GLUT isoforms expressed in the cells used in their experiments, there is a failure to investigate whether genetic Slc2a gene disruption leads to compensatory expression of other GLUT proteins. This is particularly important as the cells were sorted and expanded after lentivirus-mediated knockout of the other GLUTs prior to GLUT1 disruption. RNA-seq (or alternative method to assess for expression of each of the known GLUTs) should be done and results reported AFTER disruption of the other GLUTs.

5. Further confirmation of a lack of GLUT-mediated effects following CRISPR-mediated Slc2a disruption can be provided by assessing the effect of pharmacological GLUT inhibition, for example with cytochalasin b.

Reviewer #2: Summary: Authors utilize CRISPER-Cas 9 gene technology to ablate GLUT1 and show that 14C-glucose uptake is reduced, but that the uptake of 2-NBDG, a common fluorescent glucose that has widespread use in glucose uptake studies, is not affected. In addition, the uptake of 2-NBDG is not affected by knock out of other glucose transporters, or by ablation of select nucleoside, nucleotide, or ABC transporters. Authors conclude that 2-NBDG is taken up by cells by an unknown mechanism, but independent of glucose specificity.

Critique summary: The methodology and experimental design are appropriate, well reported and clearly described. The genetic editing technique employed in this study is a unique approach to modulate glucose transporters. The results of this study support other published work and call into question the efficacy of using 2-NBDG as a surrogate for glucose in glucose uptake studies. The data demonstrating that GLUT1 does not transport 2-NBDG is better documented than the case for the non-involvement of the other transporters studied (see below). Given the widespread use of this analog in the research literature, it is important that these results are published.

Questions and concerns:

1) I have some questions about the controls utilized in this study. The ablation of GLUT1 (Fig 1) is confirmed by both protein analysis (GLUT1 immunostaining) and by functional analysis (14C-glucose uptake). However, neither protein analysis nor functional analysis is utilized to confirm the knock out of the other putative transporters of 2-NBDG. Rather the authors rely on DNA sequencing to show the ablation of the targeted transporter and those analysis all show some unmodified targets. The actual loss of the receptor is not demonstrated. In fact, the ablation of GUT1 actually does not completely knock out GLUT1 (Fig 1), so why would we expect a complete ablation of the other transporters. I do find that data demonstrating no increase in 2NBDG negative cells convincing that the receptor was not involved in transport. (I would have expected that a certain population of cells would have both alleles ablated and thus, if that receptor were involved in transport, those cells should show up as 2NBDG negative.). Please comment on why receptor analysis and function are not reported for transporters other than GLUT1.

2) I would revise the conclusion that ‘2NBDG is actively transported’ (line 66 of introduction). ‘Active’ implies energy input required (eg ATP) for which no evidence is provided. Also, I am not entirely convinced that the data can distinguish between an actual transport process as opposed to a binding and internalization via protein recycling. The 1NBDGF control does demonstrate mediated uptake, but not the mechanism. It would have been interesting to measure uptake of just the chromophore (NBD), which I expect would be lipid soluble. The role of the fluorescent chromophore in regulating uptake is not clear.

3) Please define TPM and MFI in figure legends. Also, while stated in the figure legend, Fig 2B would be clearer if the spleen cells and bone marrow cells were designated on the figure itself (eg white print on the black photos). This is just a suggestion.

Reviewer #3: PLOS ONE

#PONE-D-21-38751 220205

Authors questioned if 2-deoxy-2[N-(7-nitrobenz-2-oxa-1,3-diazol-4-yl)amino]-D-glucose (2-NBDG), the most widely used fluorescent derivative of D-glucose, is able to monitor D-glucose uptake through glucose transporters in a mouse-derived myeloma cell line 5TGM1.

Approximately 50% loss in GLUT1-positive cells by ablation of glucose transporter gene SLC2A1 (GLUT1) by CRISPR-Cas9 produced no significant difference in the mean fluorescence intensity of cells, when cells were incubated with 2-NBDG for up to an hour in a starved (i.e., D-glucose-free) condition, whereas the radioactivity significantly reduced by the ablation compared to control when14C-labeled D-glucose was applied to cells for 30 minutes.

2-NBDG, but not NBD-fructose in which the NBD moiety was attached to the C-1 position of fructose, was transported into both splenic and bone marrow plasma cells, suggesting a specific import mechanism of 2-NBDG operates in these cells in their experimental condition.

From RNA-seq analyses, authors identified SLCA1, SLC2A3, SLC2A5, SLC2A6, SLC2A8, and SLC50A1 as candidate glucose transporters in ex vivo bone marrow plasma cells and/or in 5TGM1 cells. However, authors stated that disruption of these genes in 5TGM1 cells failed to affect the 2-NBDG uptake. None of gene mutations in SLC29 nucleoside transporters, SLC35 nucleoside-sugar transporters, and ABC transporters could prevent the 2-NBDG uptake.

Authors concluded that 2-NBDG is actively transported into cells independently of known glucose transporters, and is not a faithful indicator of glucose transport. They also mentioned that 2-NBDG should not be used as a proxy for glucose uptake by mammalian cells.

General comments:

Since D-glucose is the most fundamental energy source for living things, cells have various uptake systems for D-glucose that are not only transporters but also such as channels, endocytosis, and internalization. These divergent uptake processes may operate either simultaneously or independently, temporally and/or in a spatially localized manner in the same cells depending on the condition. The point is that it should be separately discussed to test whether 2-NBDG is imported through GLUT and to evaluate whether 2-NBDG uptake in a particular cell is affected significantly by ablation of GLUT genes. Because the latter greatly depends on the relative functional contribution of GLUT in the D-glucose transport of the cell of interest. Authors may say that they used 14C D-glucose as a control. However, see comments below.

Of course, 2-NBDG is not identical to D-glucose, as also true in major D-glucose tracers 2-DG, FDG, and 3-O-methyl-D-glucose, indicating that we should be always cautious in interpretating results obtained when using these tracers. In my opinion, this study raises an issue of importance that 2-NBDG users may encounter when evaluating cellular uptake of D-glucose by 2-NBDG, especially through high affinity glucose transporters like GLUT1.

Of particular importance when evaluating the uptake kinetics of D-glucose and of its derivatives thorough glucose transporters is that we should examine the initial uptake process. For details, see Fig. 1 in Baldwin and colleagues (J. Biol. Chem. 256: 3685-3689, 1981). As illustrated, if D- and L-glucose uptake was evaluated for 30 minutes or 60 minutes, not only 14C-labeled D-glucose but also 3H-labeled L-glucose might have been taken up considerably, suggesting that non-stereoselective, possibly non-transporter-mediated uptake of glucose had took part in this system.

For an importance of evaluating the initial uptake process, see also Fig. 1 and Fig. 2 in Johnson and colleagues (J. Biol. Chem. 265: 6548-6551, 1990). The horizontal axis of Fig. 1 in this seminal paper is in seconds. Moreover, the uptake of 3H-labeled 3-O-mythyl-D-glucose was saturated at 60 seconds and the half time of the uptake was less than 15 seconds. A similar method has been applied for evaluating 2-NBDG uptake through GLUT2 and GLUT1 (see Fig. 2 in Yamada and colleagues, J. Biol. Chem. 275: 22278-22283, 2000).

If authors would like in this paper to draw such a strong conclusion about 2-NBDG including its kinetic property, they should at least analyze the initial uptake process of 2-NBDG into their cells in a quantitative manner using a standard kinetic analysis of glucose uptake as in the references cited above. Similar experiments should be conducted for radiolabeled tracers as well for comparison.

Or, authors should state the interpretation of their results more cautiously, being aware of the limitation of their experimental procedure.

Specific comments:

1) Microscopic images of 2-NBDG uptake into cells were presented only in Figure 2A in the present study. However, the image pattern shown is atypical, because the 2-NBDG signal was detected both in the cytosolic and the nuclear compartment. Indeed, authors stated, “2-NBDG was distributed evenly across the cytosol of cells”. Usually, 2-NBDG signal is mainly localized in the cytosolic compartment that could be easily discriminated from the nuclear compartment. Authors should provide a higher resolution microscopic images for showing cellular localization of 2-NBDG with the condition used in the present study. It would be possible, since authors used an imaging flow cytometer (Imagestream Mk II, Luminex) equipped with a 60x objective lens. Then, the fluorescence intensity should be evaluated for ROIs assigned to the cytosolic compartment excluding nuclei.

2) When evaluating tumor cell lines especially when they were sub-cultured for many years, multiple non-transporter-mediated uptake processes of D-glucose may operate, or dominate in some cases, in addition to glucose transporters, even if short incubation period was used. To see details in the uptake among conditions illustrated, plot the 2-NBDG intensity in Figure 1C in a linear scale, even if it will cause a change in the shape of the background intensity profile.

3) In Figure 1D, authors combined data of the 2-NBDG uptake (the mean fluorescence intensity) for an incubation periods 15, 30, 45, and 60 minutes together. Authors should separately compare the distribution of the 2-NBDG mean fluorescence intensity for 0 minutes and 15 minutes in an expanded scale to see details more clearly in a relatively short incubation period.

4) Authors used a monoclonal antibody SPM498 (Thermo Fisher Scientific) for validating the protein expression of GLUT1 on 5TGM1 cells. However, no microscopic image for the GLUT1 expression pattern was shown. This is important. Because a membrane-spanning transporter GLUT1 should be detected on the plasma membrane of cells as shown in Figure 1 of Ogorevc et al., Biomed. Rep. 2021, https://doi.org/10.3892/br.2021.1455. In this literature, the expression of GLUT1 was evaluated by the same SPM498 and compared among human tissues, while the expression on erythrocytes present in the blood vessels inside the tissues was used as a standard to determine the staining condition. As known well, immunostaining depends critically on the antibody and methods used. Thus, it is required to show that the GLUT1 immunoreactivity is detected on the plasma membrane of these 5TGM1 cells in the immunostaining condition used. Showing the expression profiles of positive and negative control cells or tissues with the same staining condition is a minimum requirement for validating that GLUT1 expression experiment was done properly. Authors should also present specimens that show how the GLUT1 immunoreactivity is affected by the SLC2A1 gene ablation in the same staining condition.

5) In Figure 4B, although the logarithmic plot of 2-NBDG uptake somewhat obscured the difference, it appears that 2-NBDG uptake in the control gRNA is larger than Slc2a1/2/3/6/8 gRNA. Plot the 2-NBDG uptake in Figure 4B in a linear scale. Similarly, in Figure 4C, the mean fluorescence intensity of 2-NBDG is larger in control than in Slc2a1/3/5/6/8. Consistently, Figure 4D shows that %2-NBDG-negative cells appears to be larger in Slc2a1/3/5/6/8 than in control. All these data may show an effect of the gene ablation on the uptake of 2-NBDG, although authors mentioned that the difference was not significant. For the statistical analyses, authors used the Mann-Whitney non-parametric t-test. First, present a scattergram that shows the distribution of actual values for Figure 4C and 4D in supplementary information. Next indicate the number of specimens tested explicitly on the bar in Figure 4C and 4D, or in the legend. Explain the rationale why authors did not use simple paired t-test in Figure 4C and 4D?

6) For evaluating the uptake of 2-NBDG, authors used incubation period longer than 15 minutes in a starved (i.e., D-glucose-free) condition at 37°C. This may activate physiological/pathophysiological processes including internalization of proteins as well as other multiple plasma membrane transporting processes. The kinetic analysis of the initial uptake within 1 minutes would provide an opportunity to identify fast uptake separately from other relatively slow processes.

7) 2-NBDG entry may occur through an opening of GAP junction/hemichannels in some neoplastic cells as well as starved normal cells (Rouach N. et. al., Science 322: 1551-1555, 2008; Thompson, RJ. et. al., Science 312: 924-927, 2008; Gandhi, GK. et. al., J. Neurochem. 110: 857-869, 2009). Authors should also test whether carbenoxolone, a widely used GAP junction/hemichannel blocker, affects the 2-NBDG entry into 5TGM1 cells in the present experimental condition.

8) Authors used 2-NBDG of Cayman Chemical (Item No. 11046). The technical information of this item No. 11046 said that the solubility of this 2-NBDG in PBS (pH 7.2) is 10 mg/ml. However, a purified 2-NBDG is reliably dissolved in aqueous solution at a concentration of approximately 1 mg/ml due to its lipophilic moiety. In our experiments, to increase the solubility, some commercially available 2-NBDG contained a solubilizing agent that potentially affect the membrane transport properties.

9) Details for the 14C D-glucose (Perkin Elmer) used in the present study should be shown, because there are different types of 14C D-glucose in this manufacturer.

Reviewer #4: The major issue with the manuscript is the sweeping claim that 2NBDG should not be used to report on glucose uptake in mammalian cells when only one cell line is used and, to boot, the one cell line used is a plasma cell line not representative of the numerous different mammalian cell types. The authors must walk back their claim about the suitability of 2NBDG to report on glucose uptake, or repeat the experiments in the manuscript on additional cell lines (both malignant and non-malignant) that are representative of all mammalian cells. Specific comments and questions are below.

The authors claim that 2NBDG is likely taken up via a transporter based on Figure 2A and a comparison to mCherry. Why does this data suggest 2NBDG is taken up by a transporter?

How was percent positive determined in Figure 2B? It looks like not all plasma cells even take up 2NBDG based on this data, which is surprising.

In Figure 3, why was C-glucose not used to confirm knockdown of the glucose transporters affected glucose uptake? There is no appropriate control shown.

There is a discrepancy in the % positive cell data shown in Figure 2D and % negative cell data in Figure 3D. How is there well below 100% positive cells in Figure 2D but nearly 0 % negative cells in 3D?

Numerous studies have shown that D-glucose competes with and reduces uptake of 2NBDG. Competition assays are needed to demonstrate that 2NBDG is actually not a reporter on glucose uptake, which is the primary claim the authors make. It is this reviewers opinion, that all that can be said from the performed study is that the authors did not find a 2NBDG transporter in plasma cells. Wording needs to be much more specific and related to the data shown rather than a sweeping claim about 2NBDG not reporting on glucose uptake.

6. PLOS authors have the option to publish the peer review history of their article (what does this mean?). If published, this will include your full peer review and any attached files.

Reviewer #1: No

Reviewer #2: No

Reviewer #3: No

Reviewer #4: No

---

## [Author Response · Author response to Decision Letter 0]

24 May 2022

We are grateful to the editors and reviewers for their suggestions to our manuscript. They have led us to provide a more thorough characterization of 2NBDG uptake in cells and reinforces our initial hypothesis that its transport occurs independently of glucose transporters. Changes to the manuscript have been shown in purple.

Journal Requirements:

We apologize for the oversight and have formatted the manuscript to reflect the PLoS One’s style requirements and file naming.

(This work was supported by NIH grant R01AI129945 (D.B.). L. D. was supported by a Bio5 Postdoctoral fellowship award. The use of the Imagestream was made possible by the NIH award S10 OD028466. The funders had no role in study design, data collection and analysis, decision to publish, or preparation of the manuscript)

We now include a statement stating the absence of any additional external funding to this study in both the manuscript and cover letter.

We have removed instances where data has not been shown and have reworded the manuscript to reflect that.

Reviewers' comments:

Reviewer's Responses to Questions

Comments to the Author

1. Is the manuscript technically sound, and do the data support the conclusions?

Reviewer #1: Partly

Reviewer #2: Yes

Reviewer #3: Partly

Reviewer #4: Partly

2. Has the statistical analysis been performed appropriately and rigorously? 

Reviewer #1: Yes

Reviewer #2: Yes

Reviewer #3: No

Reviewer #4: Yes

3. Have the authors made all data underlying the findings in their manuscript fully available?

Reviewer #1: Yes

Reviewer #2: Yes

Reviewer #3: No

Reviewer #4: Yes

4. Is the manuscript presented in an intelligible fashion and written in standard English?

Reviewer #1: Yes

Reviewer #2: Yes

Reviewer #3: Yes

Reviewer #4: Yes

5. Review Comments to the Author

Reviewer #1: In this paper, the authors present data regarding the effect of CRISPR-mediated deletion of several classes of membrane transport proteins on 2NBDG uptake in primary plasma and 5TGM1 myeloma cell lines. The authors observe continued 2NBDG uptake despite significant reduction in glucose transport with knockdown of the primary GLUT transporter (SLC2A1) in these cells. To search for possible explanations for this unexpected finding, they perform a fairly extensive, though not exhaustive, set of experiments to knock down other transporters, including one experiment where they simultaneously knock down each of the GLUT isoforms detected in the 5TGM3 cells. While overall the data agree with recent reports using other methodologies that 2NBDG may not serve as a reliable indicator of overall glucose uptake and utilization in mammalian cells, this manuscript has several limitations, based largely on the tools utilized and the experimental conditions chosen, that require either modification of the conclusions reached or addition of additional data. In particular, while the identity of the novel putative transporter(s) may not be clear, within the context of this paper it can and should be directly established whether or not sustained 2NBDG import is due to a carrier-mediated process.

We thank this reviewer for their critique of our manuscript. Our response to their suggestions is as follows:

Specific comments and critique:

1. In the paper abstract, the authors claim that CRISP-mediated knockout of Slc2a1 alters the kinetics of 2NBDG uptake. However, the paper does not directly assess transport kinetics. The only data shown is a time course for mean fluorescence intensity. The scale shown is relatively long and does not allow assessment of zero-trans uptake. To make this claim, much more data needs to be shown including classic experiments to assess for transport-mediated uptake. This includes demonstration of saturability and at a minimum estimation of the Km and Vmax of the transport process.

New data to address this are now shown in a new figure 9A. We quantified 2NBDG uptake in cells and found that it reaches a steady state in less than 20 seconds. The maximum intensity titrates proportionately with the concentration of 2NBDG. Saturation was not reached at any of the concentrations we tested, thereby precluding calculation of Km and Vmax. It is possible that uptake of 2NBDG is mediated by a low-affinity transporter, and we have modified the text to reflect this possibility (page 24, lines 492-501). Further, ablation of GLUT1 as shown in figure 4C has no impact on the uptake kinetics of 2NBDG.

2. In Figure 1, there is a discrepancy in the degree of reduction of GLUT1 positive cells (~50%) and the degree of 14C glucose uptake (~80%) that is not adequately addressed in the manuscript text. This may be due in part to the experimental conditions used to assess for glucose transport activity. What is shown is the combined effect in GLUT1 expressing and non-expression cells with 30-minute incubation with radiolabeled substrate. Under these conditions, there are influences of both uptake and metabolism of glucose. It is recommended that assays be performed at shorter time points and with non-metabolizable substrate (e.g. 2-deoxyglucose). At a minimum, the authors need to show (or adequately reference) the kinetics of glucose transport activity in these cell lines under the conditions used.

The flow cytometric plots reflect cells that have lost both functional copies of Slc2a1, but there are additional cells that likely have lost one copy and have a partial reduction in uptake. Thus, the percentage of cells that have completely lost GLUT1 is expected to reflect a lower bound of the impact on 14C glucose uptake. We do acknowledge the simultaneous uptake and metabolism of glucose in these assays and have discussed this point in more detail in the manuscript on page 13, lines 247-252. The discrepancy between 2-deoxyglucose and 2NBDG has been addressed previously in Sinclair L et al. (1).

3. The interpretation of the data shown in Figure 2 is overstated. In these experiments, the authors seek to establish that the uptake of 2NBDG does not occur via a non-specific endocytosis-mediated process. Demonstration of uniform cytosolic staining of 2NBDG does not prove that this is mediated through a transport-mediated process. Furthermore, the comparison to 1NBDF, while showing that there is specificity for 2NBDG over the other substrate (implying an effect on glucose over fructose uptake), it does not directly follow that these data is not from GLUT1 mediated transport.

We accept this criticism and have added more nuance to the text to reflect these possibilities. Though we disfavor endocytosis as the primary mechanism (due to the absence of puncta in the imaging flow cytometry analysis in figure 2) and diffusion (due to the 2NBDG uptake kinetics in 5TGM1 cells and plasma cells in supplementary figure 3 and figure 3B), we acknowledge that these results indirectly support our conclusions. Page 15, lines 287-288 and page 16, lines 312-326 now discuss these findings with more context. 

4. Although the authors use RNA-seq to assess the specific GLUT isoforms expressed in the cells used in their experiments, there is a failure to investigate whether genetic Slc2a gene disruption leads to compensatory expression of other GLUT proteins. This is particularly important as the cells were sorted and expanded after lentivirus-mediated knockout of the other GLUTs prior to GLUT1 disruption. RNA-seq (or alternative method to assess for expression of each of the known GLUTs) should be done and results reported AFTER disruption of the other GLUTs.

New RNA-seq data to address this point are shown in supplementary figure 4. Neither deletion of Slc2a1 nor other Slc2 family members lead to an increase in expression of other Slc2 family members. Further, we see no increase in expression of Slc5a1, Slc5a2, or Slc50a1 in these datasets. 

5. Further confirmation of a lack of GLUT-mediated effects following CRISPR-mediated Slc2a disruption can be provided by assessing the effect of pharmacological GLUT inhibition, for example with cytochalasin b.

New data to address this are included in figure 4A. We observe no change in 2NBDG intensity in cultures treated with cytochalasin B. We also tested more specific GLUT1 inhibitors, WZB-117 and BAY-876 (2,3), and found similar results.

Reviewer #2: Summary: Authors utilize CRISPER-Cas 9 gene technology to ablate GLUT1 and show that 14C-glucose uptake is reduced, but that the uptake of 2-NBDG, a common fluorescent glucose that has widespread use in glucose uptake studies, is not affected. In addition, the uptake of 2-NBDG is not affected by knock out of other glucose transporters, or by ablation of select nucleoside, nucleotide, or ABC transporters. Authors conclude that 2-NBDG is taken up by cells by an unknown mechanism, but independent of glucose specificity.

Critique summary: The methodology and experimental design are appropriate, well reported and clearly described. The genetic editing technique employed in this study is a unique approach to modulate glucose transporters. The results of this study support other published work and call into question the efficacy of using 2-NBDG as a surrogate for glucose in glucose uptake studies. The data demonstrating that GLUT1 does not transport 2-NBDG is better documented than the case for the non-involvement of the other transporters studied (see below). Given the widespread use of this analog in the research literature, it is important that these results are published.

We thank the reviewer for their positive comments. The disconnect between glucose and 2NBDG uptake in our assays initially took us by surprise, but now after multiple assessments by our group and others, it is clear that this analog is not a reliable surrogate for glucose uptake in mammalian cells. Our responses to their critique are below:

Questions and concerns:

1) I have some questions about the controls utilized in this study. The ablation of GLUT1 (Fig 1) is confirmed by both protein analysis (GLUT1 immunostaining) and by functional analysis (14C-glucose uptake). However, neither protein analysis nor functional analysis is utilized to confirm the knock out of the other putative transporters of 2-NBDG. Rather the authors rely on DNA sequencing to show the ablation of the targeted transporter and those analysis all show some unmodified targets. The actual loss of the receptor is not demonstrated. In fact, the ablation of GUT1 actually does not completely knock out GLUT1 (Fig 1), so why would we expect a complete ablation of the other transporters. I do find that data demonstrating no increase in 2NBDG negative cells convincing that the receptor was not involved in transport. (I would have expected that a certain population of cells would have both alleles ablated and thus, if that receptor were involved in transport, those cells should show up as 2NBDG negative.). Please comment on why receptor analysis and function are not reported for transporters other than GLUT1.

We relied on DNA sequencing to identify frame-shift mutations in the various genes we assayed as there are no commercially available flow cytometric antibodies to reliably detect these mouse transporters. This is further complicated by even fewer radio-labelled compounds to specifically measure their function. Given that flow cytometry provides single cell resolution, we agree with the comment that we would have expected increased 2NBDG-negative cell frequencies had we even partially disrupted a functionally relevant transporter.

2) I would revise the conclusion that ‘2NBDG is actively transported’ (line 66 of introduction). ‘Active’ implies energy input required (eg ATP) for which no evidence is provided. Also, I am not entirely convinced that the data can distinguish between an actual transport process as opposed to a binding and internalization via protein recycling. The 1NBDGF control does demonstrate mediated uptake, but not the mechanism. It would have been interesting to measure uptake of just the chromophore (NBD), which I expect would be lipid soluble. The role of the fluorescent chromophore in regulating uptake is not clear.

We acknowledge this criticism and have modified the text accordingly. Line 20-21 of the abstract now indicates 2NBDG uptake to be a ‘specific mechanism unlinked from glucose transport’. We examined for uptake of the chromophore 4-Chloro-7-nitrobenzofurazan (4C7NB) and have reported it in supplementary figure 3. As pointed out by the reviewer, the dynamics of 4C7NB uptake is very different from that of 2NBDG. 

3) Please define TPM and MFI in figure legends. Also, while stated in the figure legend, Fig 2B would be clearer if the spleen cells and bone marrow cells were designated on the figure itself (eg white print on the black photos). This is just a suggestion.

We apologize for the confusion and have made the necessary changes in figures 1 for MFI (page 14, line 273) and figure 5 for TPM (page 19, line 384). TPM is short for ‘transcripts per million’ and MFI means ‘Mean fluorescent intensity’. We are not sure as to why the FACS plots showing 2NBDG uptake in splenic and bone marrow plasma cells appear to have a black background. At the time of data upload, the figures have a white background and percentage positive cells indicated in the bottom right corner.

Reviewer #3: PLOS ONE

#PONE-D-21-38751 220205

Authors questioned if 2-deoxy-2[N-(7-nitrobenz-2-oxa-1,3-diazol-4-yl)amino]-D-glucose (2-NBDG), the most widely used fluorescent derivative of D-glucose, is able to monitor D-glucose uptake through glucose transporters in a mouse-derived myeloma cell line 5TGM1.

Approximately 50% loss in GLUT1-positive cells by ablation of glucose transporter gene SLC2A1 (GLUT1) by CRISPR-Cas9 produced no significant difference in the mean fluorescence intensity of cells, when cells were incubated with 2-NBDG for up to an hour in a starved (i.e., D-glucose-free) condition, whereas the radioactivity significantly reduced by the ablation compared to control when14C-labeled D-glucose was applied to cells for 30 minutes.

2-NBDG, but not NBD-fructose in which the NBD moiety was attached to the C-1 position of fructose, was transported into both splenic and bone marrow plasma cells, suggesting a specific import mechanism of 2-NBDG operates in these cells in their experimental condition.

From RNA-seq analyses, authors identified SLCA1, SLC2A3, SLC2A5, SLC2A6, SLC2A8, and SLC50A1 as candidate glucose transporters in ex vivo bone marrow plasma cells and/or in 5TGM1 cells. However, authors stated that disruption of these genes in 5TGM1 cells failed to affect the 2-NBDG uptake. None of gene mutations in SLC29 nucleoside transporters, SLC35 nucleoside-sugar transporters, and ABC transporters could prevent the 2-NBDG uptake.

Authors concluded that 2-NBDG is actively transported into cells independently of known glucose transporters, and is not a faithful indicator of glucose transport. They also mentioned that 2-NBDG should not be used as a proxy for glucose uptake by mammalian cells.

General comments:

Since D-glucose is the most fundamental energy source for living things, cells have various uptake systems for D-glucose that are not only transporters but also such as channels, endocytosis, and internalization. These divergent uptake processes may operate either simultaneously or independently, temporally and/or in a spatially localized manner in the same cells depending on the condition. The point is that it should be separately discussed to test whether 2-NBDG is imported through GLUT and to evaluate whether 2-NBDG uptake in a particular cell is affected significantly by ablation of GLUT genes. Because the latter greatly depends on the relative functional contribution of GLUT in the D-glucose transport of the cell of interest. Authors may say that they used 14C D-glucose as a control. However, see comments below.

Of course, 2-NBDG is not identical to D-glucose, as also true in major D-glucose tracers 2-DG, FDG, and 3-O-methyl-D-glucose, indicating that we should be always cautious in interpretating results obtained when using these tracers. In my opinion, this study raises an issue of importance that 2-NBDG users may encounter when evaluating cellular uptake of D-glucose by 2-NBDG, especially through high affinity glucose transporters like GLUT1.

Of particular importance when evaluating the uptake kinetics of D-glucose and of its derivatives thorough glucose transporters is that we should examine the initial uptake process. For details, see Fig. 1 in Baldwin and colleagues (J. Biol. Chem. 256: 3685-3689, 1981). As illustrated, if D- and L-glucose uptake was evaluated for 30 minutes or 60 minutes, not only 14C-labeled D-glucose but also 3H-labeled L-glucose might have been taken up considerably, suggesting that non-stereoselective, possibly non-transporter-mediated uptake of glucose had took part in this system.

For an importance of evaluating the initial uptake process, see also Fig. 1 and Fig. 2 in Johnson and colleagues (J. Biol. Chem. 265: 6548-6551, 1990). The horizontal axis of Fig. 1 in this seminal paper is in seconds. Moreover, the uptake of 3H-labeled 3-O-mythyl-D-glucose was saturated at 60 seconds and the half time of the uptake was less than 15 seconds. A similar method has been applied for evaluating 2-NBDG uptake through GLUT2 and GLUT1 (see Fig. 2 in Yamada and colleagues, J. Biol. Chem. 275: 22278-22283, 2000).

If authors would like in this paper to draw such a strong conclusion about 2-NBDG including its kinetic property, they should at least analyze the initial uptake process of 2-NBDG into their cells in a quantitative manner using a standard kinetic analysis of glucose uptake as in the references cited above. Similar experiments should be conducted for radiolabeled tracers as well for comparison.

Or, authors should state the interpretation of their results more cautiously, being aware of the limitation of their experimental procedure.

We thank the reviewer for a detailed critique of our manuscript and for suggesting additional experiments that help improve our case against the use of 2NBDG as an analog of glucose in mammalian cells. Our responses to their suggestions are as follows:

Specific comments:

1) Microscopic images of 2-NBDG uptake into cells were presented only in Figure 2A in the present study. However, the image pattern shown is atypical, because the 2-NBDG signal was detected both in the cytosolic and the nuclear compartment. Indeed, authors stated, “2-NBDG was distributed evenly across the cytosol of cells”. Usually, 2-NBDG signal is mainly localized in the cytosolic compartment that could be easily discriminated from the nuclear compartment. Authors should provide a higher resolution microscopic images for showing cellular localization of 2-NBDG with the condition used in the present study. It would be possible, since authors used an imaging flow cytometer (Imagestream Mk II, Luminex) equipped with a 60x objective lens. Then, the fluorescence intensity should be evaluated for ROIs assigned to the cytosolic compartment excluding nuclei.

New data are shown to address this issue in Figure 2A. We have quantified colocalization of 2NBDG with bona fide cytoplasmic and nuclear stains. As pointed out by the reviewer, we do observe some colocalization with the nuclear stain and 2NBDG, but much less than is observed with the cytoplasm.

2) When evaluating tumor cell lines especially when they were sub-cultured for many years, multiple non-transporter-mediated uptake processes of D-glucose may operate, or dominate in some cases, in addition to glucose transporters, even if short incubation period was used. To see details in the uptake among conditions illustrated, plot the 2-NBDG intensity in Figure 1C in a linear scale, even if it will cause a change in the shape of the background intensity profile.

Data has been re-plotted into the linear scale.

3) In Figure 1D, authors combined data of the 2-NBDG uptake (the mean fluorescence intensity) for an incubation periods 15, 30, 45, and 60 minutes together. Authors should separately compare the distribution of the 2-NBDG mean fluorescence intensity for 0 minutes and 15 minutes in an expanded scale to see details more clearly in a relatively short incubation period.

New data to address this point are shown in figure 3B and supplementary figure 3. We examined the kinetics of 2NBDG uptake at earlier time points and observed that cells reach steady state in less than 20 seconds. Further, in figure 4C, deletion of GLUT1 in 5TGM1 cells by CRISPR-Cas9 does not affect 2NBDG uptake. 

4) Authors used a monoclonal antibody SPM498 (Thermo Fisher Scientific) for validating the protein expression of GLUT1 on 5TGM1 cells. However, no microscopic image for the GLUT1 expression pattern was shown. This is important. Because a membrane-spanning transporter GLUT1 should be detected on the plasma membrane of cells as shown in Figure 1 of Ogorevc et al., Biomed. Rep. 2021, https://doi.org/10.3892/br.2021.1455. In this literature, the expression of GLUT1 was evaluated by the same SPM498 and compared among human tissues, while the expression on erythrocytes present in the blood vessels inside the tissues was used as a standard to determine the staining condition. As known well, immunostaining depends critically on the antibody and methods used. Thus, it is required to show that the GLUT1 immunoreactivity is detected on the plasma membrane of these 5TGM1 cells in the immunostaining condition used. Showing the expression profiles of positive and negative control cells or tissues with the same staining condition is a minimum requirement for validating that GLUT1 expression experiment was done properly. Authors should also present specimens that show how the GLUT1 immunoreactivity is affected by the SLC2A1 gene ablation in the same staining condition.

New data in supplementary figure 1 shows GLUT1 staining in deleted cultures. GLUT1 localizes to the cell membrane, as demonstrated by colocalization with a surface marker CD138. This is consistent with Ogorevc M et al., Biomed. Rep. 2021. In GLUT1-negative cells in the same sample, we do not observe any surface or cytosolic GLUT1, as shown by both images and quantified by similarity morphology indices.

5) In Figure 4B, although the logarithmic plot of 2-NBDG uptake somewhat obscured the difference, it appears that 2-NBDG uptake in the control gRNA is larger than Slc2a1/2/3/6/8 gRNA. Plot the 2-NBDG uptake in Figure 4B in a linear scale. Similarly, in Figure 4C, the mean fluorescence intensity of 2-NBDG is larger in control than in Slc2a1/3/5/6/8. Consistently, Figure 4D shows that %2-NBDG-negative cells appears to be larger in Slc2a1/3/5/6/8 than in control. All these data may show an effect of the gene ablation on the uptake of 2-NBDG, although authors mentioned that the difference was not significant. For the statistical analyses, authors used the Mann-Whitney non-parametric t-test. First, present a scattergram that shows the distribution of actual values for Figure 4C and 4D in supplementary information. Next indicate the number of specimens tested explicitly on the bar in Figure 4C and 4D, or in the legend. Explain the rationale why authors did not use simple paired t-test in Figure 4C and 4D?

We have added a new graph in figure 6B in the linear scale. As indicated in the figure legend, the data and statistics are pooled from three experiments. We have carried out a paired t-test for figure 6C and D as suggested. P values observed for figure 6C is 0.1298 and for 6D is 0.0843 and have been indicated on the figure.

6) For evaluating the uptake of 2-NBDG, authors used incubation period longer than 15 minutes in a starved (i.e., D-glucose-free) condition at 37°C. This may activate physiological/pathophysiological processes including internalization of proteins as well as other multiple plasma membrane transporting processes. The kinetic analysis of the initial uptake within 1 minutes would provide an opportunity to identify fast uptake separately from other relatively slow processes.

As indicated in the materials and methods, we carried out all 2NBDG assays (unless indicated otherwise) in complete RPMI supplemented with fetal bovine serum and as such is a glucose-sufficient environment. We have now quantified 2NBDG uptake in a glucose-free environment in figure 7A and observe no impact of excess competing D-glucose in a 60 second window. 

7) 2-NBDG entry may occur through an opening of GAP junction/hemichannels in some neoplastic cells as well as starved normal cells (Rouach N. et. al., Science 322: 1551-1555, 2008; Thompson, RJ. et. al., Science 312: 924-927, 2008; Gandhi, GK. et. al., J. Neurochem. 110: 857-869, 2009). Authors should also test whether carbenoxolone, a widely used GAP junction/hemichannel blocker, affects the 2-NBDG entry into 5TGM1 cells in the present experimental condition.

New data to assess the role of carbenoxolone on 2NBDG uptake are shown in figure 4B. Carbenoxolone does not impact 2NBDG uptake.

8) Authors used 2-NBDG of Cayman Chemical (Item No. 11046). The technical information of this item No. 11046 said that the solubility of this 2-NBDG in PBS (pH 7.2) is 10 mg/ml. However, a purified 2-NBDG is reliably dissolved in aqueous solution at a concentration of approximately 1 mg/ml due to its lipophilic moiety. In our experiments, to increase the solubility, some commercially available 2-NBDG contained a solubilizing agent that potentially affect the membrane transport properties.

As pointed out by the reviewer, 2NBDG from Cayman chemical is sold as a crystalline solid. We resuspend this compound to an initial concentration of 10mg/mL in 1x PBS and further aliquot it to 1mg/mL in 1x PBS. We observe complete solubility of the compound at both concentrations and do not add any solubilizing agent to any of the preparations. We have provided this updated description to the materials and methods section on page 6, lines 105-108.

9) Details for the 14C D-glucose (Perkin Elmer) used in the present study should be shown, because there are different types of 14C D-glucose in this manufacturer.

NEC042X as manufactured by Perkin Elmer is a uniformly 14C labelled glucose compound that is sold in three volumes. Each have the same catalog number but with an addendum to the existing alphanumeric code to reflect the volume. We have updated the materials and methods section to reflect our purchase of NEC042X050UC, the 50μCi preparation and added its specific activity, which is 275 mCi/mmol. Page 10, line 187 details this information.

Reviewer #4: The major issue with the manuscript is the sweeping claim that 2NBDG should not be used to report on glucose uptake in mammalian cells when only one cell line is used and, to boot, the one cell line used is a plasma cell line not representative of the numerous different mammalian cell types. The authors must walk back their claim about the suitability of 2NBDG to report on glucose uptake, or repeat the experiments in the manuscript on additional cell lines (both malignant and non-malignant) that are representative of all mammalian cells. Specific comments and questions are below.

We have added new data on primary plasma cells, B cells, and total spleen cells demonstrating that large excesses of competing glucose fail to inhibit 2NBDG uptake in figure 7. While we are uncertain what the reviewer means by ‘representative of all mammalian cells’, we refer to three other manuscripts that have also observed a discrepancy between 2NBDG and glucose uptake. First, the observations of Hamilton K et al., who have examined 2NBDG uptake in the mouse fibroblast cell line L929 and second by Sinclair L et al. who have found similar discrepancies in mouse primary T cells, and Reinfeld B et al., who have found discrepancies between FDG and 2NBDG uptake in vivo (1,4,5). We discuss these manuscripts in the discussion, and they serve as orthogonal confirmations of the experiments we have carried out in our manuscript. 

The authors claim that 2NBDG is likely taken up via a transporter based on Figure 2A and a comparison to mCherry. Why does this data suggest 2NBDG is taken up by a transporter?

As a hydrophilic compound, uptake of 2NBDG is presumably mediated either by a transporter which delivers the compound to the cytoplasm or via endocytosis, in which the compound would be expected to localize preferentially to vesicles. Our data would indicate that there is mediated uptake, both by colocalization data in figure 2 and the presence of 2NBDG negative plasma cells in 2NBDG-injected mice in figure 3A. We have edited the text to reflect these observations on page 15, lines 279-290, page 16 lines 306-323 and page 17, lines 324-326.

How was percent positive determined in Figure 2B? It looks like not all plasma cells even take up 2NBDG based on this data, which is surprising.

We note two contours in the upper left panel of figure 3A indicative of two populations that differ in their intensity of 2NBDG (x-axis). We gate on the population with higher 2NBDG intensity as 2NBDG-positive cells and confirm this by using the same gate on a different sample of plasma cells obtained from mice not injected with 2NBDG. We refer the reviewer to our previous findings demonstrating that long lived plasma cells take up larger amounts of 2NBDG in vivo than do short-lived cells (6,7). 

In Figure 3, why was C-glucose not used to confirm knockdown of the glucose transporters affected glucose uptake? There is no appropriate control shown.

To be clear, these experiments are not knockdown/shRNA assays. They are CRISPR-mediated indel mutations that create early frameshifts, validated by multiple guide RNAs targeting different regions in the gene. Null mutations can thus be easily quantified using next generation sequencing. We observed in figure 1B that ablation of GLUT1 in 5TGM1 cells induced approximately 80% reduction in 14C-glucose uptake. As shown in what is now figure 5A, most of the other Slc2 family members are expressed at quite low levels in 5TGM1 cells and are thought to preferentially mediate uptake of other sugars. Detecting further loss of glucose uptake beyond what is already seen in the absence of GLUT1 would be very challenging. We do demonstrate a high frequency of frameshift mutations in the genomic sequences of the assayed glucose transporters figure 5B. Despite this we see no impact on 2NBDG uptake.

There is a discrepancy in the % positive cell data shown in Figure 2D and % negative cell data in Figure 3D. How is there well below 100% positive cells in Figure 2D but nearly 0 % negative cells in 3D?

In what is now figure 3A, we report the frequencies of percent 2NBDG positive cells in primary plasma cells from mouse spleens and bone marrows. In what is now figure 5D, we are examining frequencies of 2NBDG negative 5TGM1 cells that have been deleted for the various Slc2 family members and Slc50a1. As mentioned above, this distribution is expected for primary plasma cells.

Numerous studies have shown that D-glucose competes with and reduces uptake of 2NBDG. Competition assays are needed to demonstrate that 2NBDG is actually not a reporter on glucose uptake, which is the primary claim the authors make. It is this reviewers opinion, that all that can be said from the performed study is that the authors did not find a 2NBDG transporter in plasma cells. Wording needs to be much more specific and related to the data shown rather than a sweeping claim about 2NBDG not reporting on glucose uptake.

New data indicating the influence of competing glucose on 2NBDG uptake is shown in now figure 7A. We do not observe any change in the kinetics of 2NBDG uptake in the presence of titrating amounts of glucose. While not statistically significant, if anything, 30mM and 10mM glucose in the assay medium showed higher intensity earlier than control cells in glucose-free media. We acknowledge that we were unable to find the putative transporter for 2NBDG through our experiments. However, we do provide ample evidence now that 2NBDG uptake is distinct from glucose uptake and should be treated as such. Had earlier studies provided this level of rigor, we very much doubt that this reagent would be in as widespread use as a surrogate for glucose uptake.

References

1. Sinclair LV, Barthelemy C, Cantrell DA. Single Cell Glucose Uptake Assays: A Cautionary Tale. Immunometabolism. 2020 Aug 17;2(4):e200029. 

2. Siebeneicher H, Cleve A, Rehwinkel H, Neuhaus R, Heisler I, Müller T, et al. Identification and Optimization of the First Highly Selective GLUT1 Inhibitor BAY-876. ChemMedChem. 2016;11(20):2261–71. 

3. Ojelabi OA, Lloyd KP, Simon AH, De Zutter JK, Carruthers A. WZB117 (2-Fluoro-6-(m-hydroxybenzoyloxy) Phenyl m-Hydroxybenzoate) Inhibits GLUT1-mediated Sugar Transport by Binding Reversibly at the Exofacial Sugar Binding Site*. Journal of Biological Chemistry. 2016 Dec 1;291(52):26762–72. 

4. Hamilton KE, Bouwer MF, Louters LL, Looyenga BD. Cellular binding and uptake of fluorescent glucose analogs 2-NBDG and 6-NBDG occurs independent of membrane glucose transporters. Biochimie. 2021 Nov 1;190:1–11. 

5. Reinfeld BI, Madden MZ, Wolf MM, Chytil A, Bader JE, Patterson AR, et al. Cell-programmed nutrient partitioning in the tumour microenvironment. Nature. 2021 May;593(7858):282–8. 

6. Lam WY, Becker AM, Kennerly KM, Wong R, Curtis JD, Llufrio EM, et al. Mitochondrial Pyruvate Import Promotes Long-Term Survival of Antibody-Secreting Plasma Cells. Immunity. 2016 Jul 19;45(1):60–73. 

7. Lam WY, Jash A, Yao CH, D’Souza L, Wong R, Nunley RM, et al. Metabolic and Transcriptional Modules Independently Diversify Plasma Cell Lifespan and Function. Cell Rep. 2018 Aug 28;24(9):2479-2492.e6.

---

## [Decision Letter · Decision Letter 1]

1 Jul 2022

PONE-D-21-38751R1Genetic evidence that uptake of the fluorescent analog 2NBDG occurs independently of known glucose transportersPLOS ONE

Dear Dr. Bhattacharya,

Thank you for submitting your manuscript to PLOS ONE. After careful consideration, we feel that it has merit but does not fully meet PLOS ONE’s publication criteria as it currently stands. Therefore, we invite you to submit a revised version of the manuscript that addresses the points raised during the review process. Your manuscript can be accepted after minor revision of the wordings suggested by one of the reviewers.

We look forward to receiving your revised manuscript.

Kind regards,

Hodaka Fujii, M.D., Ph.D.

Academic Editor

PLOS ONE

Journal Requirements:

Reviewers' comments:

Reviewer's Responses to Questions

**Comments to the Author**

1. If the authors have adequately addressed your comments raised in a previous round of review and you feel that this manuscript is now acceptable for publication, you may indicate that here to bypass the “Comments to the Author” section, enter your conflict of interest statement in the “Confidential to Editor” section, and submit your "Accept" recommendation.

Reviewer #2: All comments have been addressed

Reviewer #3: (No Response)

2. Is the manuscript technically sound, and do the data support the conclusions?

Reviewer #2: (No Response)

Reviewer #3: Yes

3. Has the statistical analysis been performed appropriately and rigorously? 

Reviewer #2: (No Response)

Reviewer #3: Yes

4. Have the authors made all data underlying the findings in their manuscript fully available?

Reviewer #2: (No Response)

Reviewer #3: Yes

5. Is the manuscript presented in an intelligible fashion and written in standard English?

Reviewer #2: (No Response)

Reviewer #3: Yes

6. Review Comments to the Author

Reviewer #2: (No Response)

Reviewer #3: Authors have made a considerable revision to the original manuscript. My concern is that they generalized results obtained from a limited types of cells expressing GLUT1 as "Thus, cellular uptake of 2NBDG is not a faithful indicator of glucose transport ...” (the last sentence in Abstract). I would strongly recommend authors to confine their statement to glucose transport at least through GLUT1, since no experiment was done for cells intrinsically expressing such as GLUT2.

7. PLOS authors have the option to publish the peer review history of their article (what does this mean?). If published, this will include your full peer review and any attached files.

Reviewer #2: No

Reviewer #3: No

---

## [Author Response · Author response to Decision Letter 1]

1 Jul 2022

Journal Requirements:

We have examined all the references and found them to be complete and correct. All cited manuscripts are currently accepted and available on the publisher’s website or on PubMed Central.

Reviewers' comments:

Reviewer's Responses to Questions

Comments to the Author

1. If the authors have adequately addressed your comments raised in a previous round of review and you feel that this manuscript is now acceptable for publication, you may indicate that here to bypass the “Comments to the Author” section, enter your conflict of interest statement in the “Confidential to Editor” section, and submit your "Accept" recommendation.

Reviewer #2: All comments have been addressed

Reviewer #3: (No Response)

2. Is the manuscript technically sound, and do the data support the conclusions?

Reviewer #2: (No Response)

Reviewer #3: Yes

3. Has the statistical analysis been performed appropriately and rigorously? 

Reviewer #2: (No Response)

Reviewer #3: Yes

4. Have the authors made all data underlying the findings in their manuscript fully available?

Reviewer #2: (No Response)

Reviewer #3: Yes

5. Is the manuscript presented in an intelligible fashion and written in standard English?

Reviewer #2: (No Response)

Reviewer #3: Yes

6. Review Comments to the Author

Reviewer #2: (No Response)

Reviewer #3: Authors have made a considerable revision to the original manuscript. My concern is that they generalized results obtained from a limited types of cells expressing GLUT1 as "Thus, cellular uptake of 2NBDG is not a faithful indicator of glucose transport ...” (the last sentence in Abstract). I would strongly recommend authors to confine their statement to glucose transport at least through GLUT1, since no experiment was done for cells intrinsically expressing such as GLUT2.

We apologize for our obstinacy, but even for GLUT1, we cannot conclude with certainty that 2NBDG cannot be transported through it at all. Instead, the data show that there are other glucose transporter-independent pathways by which 2NBDG predominantly enters cells. This background is what makes 2NBDG unreliable as a surrogate for glucose uptake. Whether GLUT2 or other glucose transporters are expressed does not change this fact. In this sense, we feel that it would be an overreach to adopt this suggestion and claim 2NBDG cannot be transported by GLUT1. 

7. PLOS authors have the option to publish the peer review history of their article (what does this mean?). If published, this will include your full peer review and any attached files.

Do you want your identity to be public for this peer review? For information about this choice, including consent withdrawal, please see our Privacy Policy.

Reviewer #2: No

Reviewer #3: No

---

## [Decision Letter · Decision Letter 2]

1 Aug 2022

PONE-D-21-38751R2Genetic evidence that uptake of the fluorescent analog 2NBDG occurs independently of known glucose transportersPLOS ONE

Dear Dr. Bhattacharya,

Thank you for submitting your manuscript to PLOS ONE. After careful consideration, we feel that it has merit but does not fully meet PLOS ONE’s publication criteria as it currently stands. Therefore, we invite you to submit a revised version of the manuscript that addresses the points raised during the review process.

 One of the reviewers still raised some concerns about the wordings of your revised manuscript. In this regard, I don't see any problem in the wordings of the title because your data showed that there exist some mechanisms of uptake of 2NBDG independent of known glucose transporters. On the other hand, the last sentences of the Abstract and Discussion might be very strong statements. Can you consider making their expression a little milder? For example, the last sentence of the Abstract can be "Thus, cellular uptake of 2NBDG is not necessarily a faithful indicator of glucose transport...", and the last sentence of the Discussion can be "...., we advise that despite their convenience, it should not be considered as direct indicator of glucose uptake." I think that researchers can use the 2NBDG uptake assay but should carefully interpret their results.

We look forward to receiving your revised manuscript.

Kind regards,

Hodaka Fujii, M.D., Ph.D.

Academic Editor

PLOS ONE

Journal Requirements:

Reviewers' comments:

Reviewer's Responses to Questions

**Comments to the Author**

1. If the authors have adequately addressed your comments raised in a previous round of review and you feel that this manuscript is now acceptable for publication, you may indicate that here to bypass the “Comments to the Author” section, enter your conflict of interest statement in the “Confidential to Editor” section, and submit your "Accept" recommendation.

Reviewer #3: All comments have been addressed

2. Is the manuscript technically sound, and do the data support the conclusions?

Reviewer #3: Partly

3. Has the statistical analysis been performed appropriately and rigorously? 

Reviewer #3: Yes

4. Have the authors made all data underlying the findings in their manuscript fully available?

Reviewer #3: Yes

5. Is the manuscript presented in an intelligible fashion and written in standard English?

Reviewer #3: Yes

6. Review Comments to the Author

Reviewer #3: This reviewer agrees an involvement of non-GLUT1-mediated mechanism in the uptake of 2-NBDG into certain cell types. In this respect, this manuscript provides valuable information to the scientific community. However, this reviewer unfortunately can not accept authors' conclusion and the title in the present form. Authors' evidence was obtained from limited cell types and was limited to GLUT1. However, the title of the manuscript included sentence "uptake of the fluorescent analog 2NBDG occurs independently of 'known' glucose transporters". This title would not be adequate, if 2-NBDG uptake occurs through GLUT2 in other cell lines to a significant extent. To greatly increase values of this important manuscript, this reviewer recommends authors to refine the title and the conclusive sentence, or at least to mention GLUT2.

7. PLOS authors have the option to publish the peer review history of their article (what does this mean?). If published, this will include your full peer review and any attached files.

Reviewer #3: No

---

## [Author Response · Author response to Decision Letter 2]

2 Aug 2022

Reviewer #3: This reviewer agrees an involvement of non-GLUT1-mediated mechanism in the uptake of 2-NBDG into certain cell types. In this respect, this manuscript provides valuable information to the scientific community. However, this reviewer unfortunately can not accept authors' conclusion and the title in the present form. Authors' evidence was obtained from limited cell types and was limited to GLUT1. However, the title of the manuscript included sentence "uptake of the fluorescent analog 2NBDG occurs independently of 'known' glucose transporters". This title would not be adequate, if 2-NBDG uptake occurs through GLUT2 in other cell lines to a significant extent. To greatly increase values of this important manuscript, this reviewer recommends authors to refine the title and the conclusive sentence, or at least to mention GLUT2.

As suggested by the editor, we have modified the abstract to now end with the statement, “Thus, cellular uptake of 2NBDG is not necessarily a faithful indicator of glucose transport and is promoted by an unknown mechanism.” Further, the discussion now concludes the manuscript with the statement, “… we strongly advise that despite their convenience, these assays should not be considered as direct indicators of glucose uptake.”

---

## [Editor Report · Decision Letter 3]

11 Aug 2022

Genetic evidence that uptake of the fluorescent analog 2NBDG occurs independently of known glucose transporters

PONE-D-21-38751R3

Dear Dr. Bhattacharya,

We’re pleased to inform you that your manuscript has been judged scientifically suitable for publication and will be formally accepted for publication once it meets all outstanding technical requirements.

Kind regards,

Hodaka Fujii, M.D., Ph.D.

Academic Editor

PLOS ONE
---

## [Editor Report · Acceptance letter]

15 Aug 2022

PONE-D-21-38751R3 

Genetic evidence that uptake of the fluorescent analog 2NBDG occurs independently of known glucose transporters 

Dear Dr. Bhattacharya:

I'm pleased to inform you that your manuscript has been deemed suitable for publication in PLOS ONE. Congratulations! Your manuscript is now with our production department. 

Kind regards, 

on behalf of

Dr. Hodaka Fujii 

Academic Editor

PLOS ONE